# HINT MARGINALIZATION FOR IMPROVED REASONING IN LARGE LANGUAGE MODELS

## ABSTRACT

Large Language Models (LLMs) have exhibited an impressive capability to perform reasoning tasks, especially if they are encouraged to generate a sequence of intermediate steps. Reasoning performance can be improved by suitably combining multiple LLM responses, generated either in parallel in a single query, or via sequential interactions with LLMs throughout the reasoning process. Existing strategies for combination, such as self-consistency and progressive-hint-prompting, make inefficient usage of the LLM responses. We present Hint Marginalization, a novel and principled algorithmic framework to enhance the reasoning capabilities of LLMs. Our approach can be viewed as an iterative sampling strategy for forming a Monte Carlo approximation of an underlying distribution of answers, with the goal of identifying the mode — the most likely answer. Empirical evaluation on several benchmark datasets for arithmetic reasoning demonstrates the superiority of the proposed approach.

## 1 INTRODUCTION

As Large Language Models (LLMs) have increased in size, they have demonstrated increasing reasoning abilities (Brown et al., 2020), despite not being explicitly trained to reason (Wei et al., 2022a). In particular, Chain-of-Thought (CoT) prompting has become standard for eliciting these abilities, either through few-shot examples (Wei et al., 2022b) or via a triggering sentence such as "Let's think step by step" (Kojima et al., 2022). Nevertheless, although LLMs often produce correct reasoning steps, they struggle with higher-level planning (Saparov and He, 2022), motivating researchers to explore strategies to remedy this deficiency. An effective solution is to sample several chains-of-thoughts and take the most common answer as the final vote, an approach called Self-Consistency (CoT+SC) (Wang et al., 2023). However, despite its impressive empirical performance, the gains quickly plateau on many benchmarks, often with no improvement after five samples (Aggarwal et al., 2023). Thus, a more complex reasoning strategy appears necessary.

One promising direction involves encouraging LLMs to iteratively refine their reasoning, like humans often do (Shinn et al., 2023; Li et al., 2023; Gou et al., 2023; Madaan et al., 2023). However, Huang et al. (2024) demonstrate that the capability and effectiveness of LLMs' self-correction is largely overstated in the existing literature due to the use of oracle labels for determining stopping criteria (Shinn et al., 2023), unfair experimental protocols (Du et al., 2023), and sub-optimal initial prompt design (Madaan et al., 2023). Moreover, the review/feedback prompts employed in these approaches are often long and complex, and include detailed, intricate, hand-crafted examples, tailored for specific domains or benchmarks. In spite of such extensive prompt engineering, Huang et al. (2024) observe that most of these approaches perform worse than self-consistency in a fair evaluation setting.

In this paper, we propose a novel iterative strategy called *Hint Marginalization (HM)* that offers a more principled and practical way of reasoning with refinement. We consider a setting where we can conduct LLM calls sequentially or in parallel. Our method does not make major changes to the initial prompt in later calls, and we do not need extensive prompting effort to invoke an LLM's review of the previous answers. The process starts by constructing an initial distribution of answers using CoT. In subsequent rounds, we present the unique answers from the previous round to the LLM as hints in the prompt. This leads to a new collection of answers, which we use to refine the answer distribution via a marginalization process. By maintaining a distribution and performing marginalization, we reduce sampling variance and make more efficient usage of the LLM calls.

We make the following contributions:

- We introduce a novel iterative hint-based refinement strategy for reasoning with LLMs, with the key differentiator that the method maintains and updates a *distribution* over answers. Our work highlights that an LLM can indeed derive benefit from self-reflecting on distributions of its past answers when attempting arithmetic reasoning tasks, without a need to resort to extensive prompt design or hand-crafted examples.
- Via multiple experiments with GPT-3.5 Turbo (Brown et al., 2020), GPT-4 Turbo (OpenAI et al., 2024), and the cost efficient GPT-4o-mini, we show that Hint Marginalization leads to improved arithmetic reasoning performance compared to *state-of-the-art* baselines for the same number of LLM calls and comparable token cost. We conduct experiments carefully to ensure there is no evaluation bias in favour of methods that employ refinement. Notably, out of 18 experimental scenarios (3 LLMs, six datasets), we observe a statistically significant increase in accuracy in 14.
- We show in the experiments that previous hint-based strategies such as Progressive-Hint-Prompting (PHP) Zheng et al. (2023) can be combined successfully with our method. In experiments that ensure fair evaluation, PHP combined with Self Consistency (PHP+SC) often struggles to outperform the much simpler SC. In contrast, when combined with our approach, the resultant PHP+HM procedure achieves statistically significant accuracy improvements in most experimental settings.

## 2 PROBLEM STATEMENT

Let $x$ be a question or a task in natural language, described in one or more sentences. Its true answer is denoted $y$, which can take different forms depending on the context, such as a number, a True/False boolean variable, or an option (a)/(b)/(c) from a multiple-choice set. Potentially, we have access to a (small) set of triplets $\boldsymbol{I} = \{(x_j, z_j, y_j)\}_{j=1}^K$ corresponding to semantically-similar questions $x_j$, answers $y_j$, and rationales $z_j$. Each rationale $z_j$ is a sequence of short sentences that describe the step-by-step reasoning process leading to the answer $y_j$.

We assume that we can query the LLM in series or in parallel. Our task is to design a strategy for prompting the LLM and combining the responses to provide an answer $\hat{y}$ for the question $x$. Performance is measured in terms of the average accuracy of the response, i.e., $\mathbb{E}[\mathbf{1}(\hat{y} = y)]$ for the indicator function $\mathbf{1}$.

## 3 METHODOLOGY

When presented with the question, an LLM produces a random answer $\tilde{y}$, drawn from an internal distribution that is dependent on the prompt and the LLM's parameters. To avoid notational clutter, we suppress these dependencies and denote this distribution by $p(\tilde{y}|x)$. This distribution is analytically intractable but one can sample from it directly by prompting the LLM and subsequently collecting its answer.

The reasoning ability of the LLM, i.e., the probability of producing the correct answer, is improved by careful construction of the prompt. For example, an encouragement to produce an explanation/rationale in the form of a sequence of short sentences to describe the step-by-step reasoning process has been shown to ameliorate LLMs' performance significantly compared to direct prompting (Wei et al., 2022b). We denote the provided rationale as $z$, so the response of the LLM is a pair $(z, \tilde{y})$. If rationale-annotated in-context examples are available, then reasoning can be improved by incorporating in the prompt a (small) set in the form of triplets $\boldsymbol{I} = \{(x_j, z_j, y_j)\}_{j=1}^K$.

Viewing the LLM response as a sample from the distribution, we can hypothesize that, if the LLM is capable of effective reasoning for the presented question, the mode of the distribution is most likely to be the correct answer. We would therefore like to extract the mode. One approach is to sample, either in parallel or sequentially, multiple LLM responses (each containing a rationale and answer). We can then select the answer corresponding to the Monte Carlo estimate of the mode by taking a majority vote over the sampled responses (Wang et al., 2023).

It has been observed that LLM output can be improved via a refinement or self-reflection process (Zheng et al., 2023; Wu et al., 2024; Li et al., 2023; Madaan et al., 2023; Park et al., 2023). In this process, the LLM is provided with its previous response, and asked to take it into account, or

criticize it, before producing a refined response. One form of refinement is to provide the previous answer as a hint ("The answer is close to" or "The answer is probably").

This observation is the cornerstone of our proposed methodology. Rather than seeking the mode of the original distribution $p(\tilde{y}|x)$, we construct a sequence of distributions $\{p_r(\tilde{y}|x)\}_{r>1}$, where each successive distribution in the sequence is constructed via a refinement process using samples from the previous distribution, as the number of interactions with the LLM, $r$, grows. This refinement process involves marginalization over the LLM's previous answers, which are used as hints in the current iteration. We initialize $p_1(\tilde{y}|x) = p(\tilde{y}|x)$, i.e., we start with the distribution of answers obtained from the first interaction with the LLM at $r = 1$. Our hypothesis is that the probability of the correct answer, $p_r(y|x)$, increases with $r$, so the mode of a distribution later in the sequence, i.e., $r > 1$, is more likely to be correct than the mode of $p(\tilde{y}|x)$.

### 3.1 INTUITION

We now provide an example to illustrate why marginalizing over hints should make the mode of the inference distribution more likely to be the correct answer. Suppose that we are presented with a binary question $x$ with answer, say, $y = 1$, and let us say that the probability of the correct answer, with no hints, is initially relatively low, $p(\tilde{y}{=}1|x) = 0.4$. However, when we provide the correct answer as a hint, the LLM is much more likely to answer correctly, $p(\tilde{y}{=}1|x, \mathrm{Hint}(y{=}1)) = 0.8$. Hinting at the incorrect answer 0 also strongly tilts the LLM towards that answer, but crucially, with slightly less probability, $p(\tilde{y}{=}0|x, \mathrm{Hint}(y{=}0)) = 0.6$. This is not unexpected or unusual, because it is often easier to see the truth of a statement in hindsight (or verify rather than solve unaided). With our proposed hint marginalization procedure, the updated distribution of the answer would be:

$$
\begin{aligned}
p_2(\tilde{y}{=}1|x) &= p_1(\tilde{y}{=}1|x)p(\tilde{y}{=}1|x, \mathrm{Hint}(y{=}1)) + p_1(\tilde{y}{=}0|x)p(\tilde{y}{=}1|x, \mathrm{Hint}(y{=}0)) \\
&= 0.4 \times 0.8 + (1 - 0.4) \times (1 - 0.6) \\
&= 0.56 > 0.4.
\end{aligned}
$$

Not only is the probability higher than before, but crucially, the mode of the distribution now aligns with the right answer ($y = 1$).

More generally, this augmentation will be observed whenever the flow of probability mass into the correct answer exceeds the flow of probability mass out of the correct answer. The flow out is $p_1(\tilde{y}{=}y|x)(1 - p(\tilde{y}{=}y|x, \mathrm{Hint}(y)))$, whereas the flow in is $\sum_{y' \neq y} p_1(\tilde{y}{=}y'|x)p(\tilde{y}{=}y|x, \mathrm{Hint}(y'))$. Since we expect $p_1(\tilde{y}{=}y|x, \mathrm{Hint}(y)))$ to be close to 1, the flow out is likely to be small. By contrast, we might anticipate that when the LLM is presented with an incorrect hint, it can often ignore it to a large extent. Let us assume that $p(\tilde{y}{=}y|x, \mathrm{Hint}(y')) > cp_1(\tilde{y}{=}y|x)$ for all $y'$ for some positive constant $c < 1$. Then the flow in exceeds $cp_1(\tilde{y}{=}y|x)(1 - p_1(\tilde{y}{=}y|x))$. Thus, if $p(\tilde{y}{=}y|x, \mathrm{Hint}(y))) > 1 - c(1 - p_1(\tilde{y}{=}y|x))$, the mass assigned to the correct answer will increase. For example, consider $p_1(\tilde{y}{=}y|x)) = 0.4$ and $c = 0.3$. Then we need $p(\tilde{y}{=}y|x, \mathrm{Hint}(y))) > 1 - 0.3 \times (1 - 0.4) = 0.82$.

We also note that repeated application of this procedure is further advantageous, which motivates the iterative version of our algorithm. We formalize this intuition into a general procedure and provide an algorithm for approximating these distributions next.

### 3.2 HINT MARGINALIZATION

Our approach, as the name suggests, updates the distribution of answers via marginalization over the answers obtained at the previous iteration. We denote the conditional probability of yielding $\tilde{y}$ as the answer for the task $x$ with a hint $y'$ by $p(\tilde{y}|x, \mathrm{Hint}(y'))$. We define a sequence of distributions $\{p_r(\tilde{y}|x)\}_{r>1}$, where two successive distributions are related as follows:

$$
p_{r+1}(\tilde{y}|x) = \int p(\tilde{y}|x, \mathrm{Hint}(y'))p_r(y'|x)\, dy' . \tag{1}
$$

The integral is replaced by a sum when $\tilde{y}$ is discrete, e.g., for multiple-choice questions.

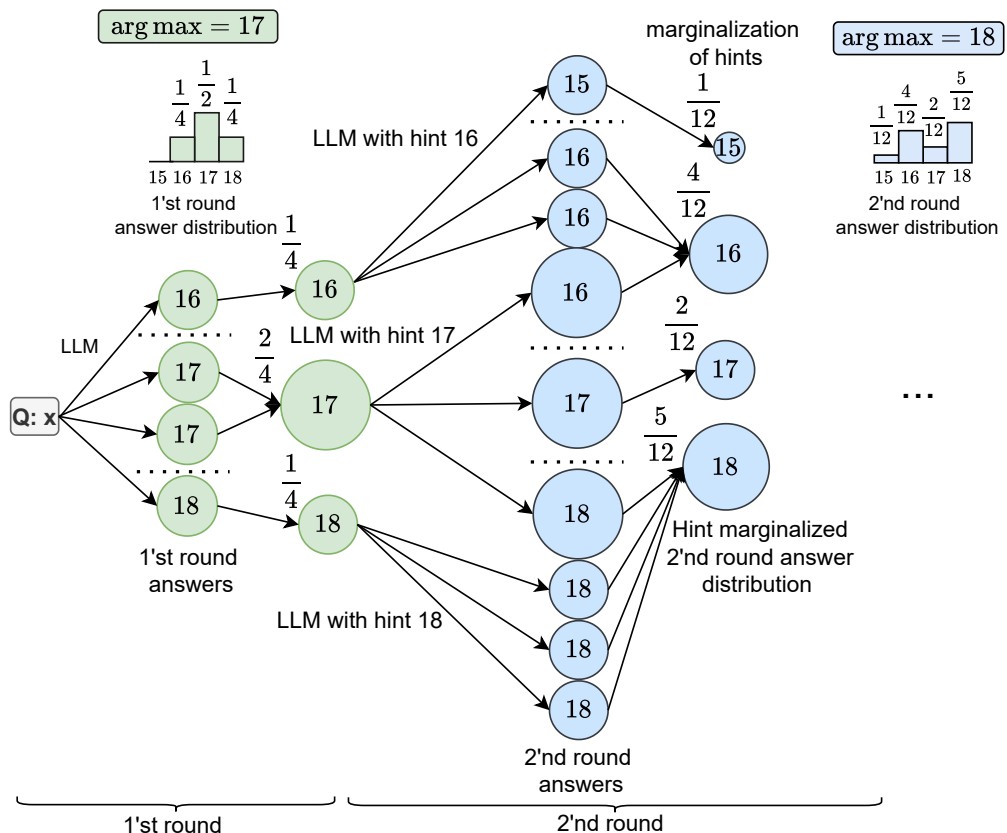

Figure 1: Illustration of one iteration of our proposed method, Hint Marginalization (HM). At its initialization, a distribution of answers is obtained from the LLM via multiple queries. In each subsequent iteration, new answers are sampled by providing each distinct old answer as a hint in the prompt. The resulting samples are then accordingly weighted by the probability of the hints for marginalization.

## 3.3 IMPLEMENTATION

We now outline the steps for performing one iteration of Hint Marginalization. As a concrete example, Figure 1 illustrates the procedure of approximating $p_2(\tilde{y}|x)$ from $p_1(\tilde{y}|x)$ in detail. Since neither $p(\tilde{y}|x, \text{Hint}(y'))$ nor $p_r(\tilde{y}|x)$ can be computed analytically, we need to resort to a Monte Carlo approach for estimating $p_{r+1}(\tilde{y}|x)$.

Suppose, at the end of the $r$-th iteration, $p_r(\tilde{y}|x)$ is approximated as follows:

$$p_r(\tilde{y}|x) \approx \sum_{m=1}^{M} \omega^m \delta(\tilde{y} - y^m), \tag{2}$$

where $\delta(\cdot)$ is the Kronecker delta function, $\{y^m\}_{m=1}^{M}$ is the set of distinct answers, and $\omega^m$ is the estimated probability of obtaining the answer $y^m$ under the distribution $p_r(\cdot|x)$. For example, in Figure 1, at $r = 1$, we have $M = 3$ distinct answers $y^1 = 16, y^2 = 17$, and $y^3 = 18$, with estimated probabilities $\omega^1 = \frac{1}{4}, \omega^2 = \frac{1}{2}$, and $\omega^3 = \frac{1}{4}$ respectively. If the correct answer $y = 18$, then the LLM's current answer $\hat{y} = 17$, based on the estimated mode of $p_1(\tilde{y}|x)$, is incorrect.

Assuming a sampling budget of $B_{r+1}$, which denotes the maximally allowed number of answers to be sampled at the $(r+1)$-th iteration, we modify, for each $m = 1, \ldots, M$, the prompt by appending $y^m$ as the hint, and sample $\lfloor \frac{B_{r+1}}{M} \rfloor$ answers subsequently. This forms the following Monte Carlo

approximation:

$$p(\tilde{y}|x, \text{Hint}(y = y^m)) \approx \sum_{\ell=1}^{L_m} \bar{\omega}^{\ell,m} \delta(\tilde{y} - y^{\ell,m}). \tag{3}$$

Here, $\{y^{\ell,m}\}_{\ell=1}^{L_m}$ are the $L_m$ distinct answers extracted from the $\lfloor \frac{B_{r+1}}{M} \rfloor$ answers and $\bar{\omega}^{\ell,m}$ is the estimated probability of having $y^{\ell,m}$ as the answer conditioned on the hint $y^m$. In Figure 1, the total budget for $r = 2$, i.e. $B_2 = 9$, the number of distinct answers for different hints are $L_1 = 2$, $L_2 = 3$, and $L_3 = 1$. For the hint $y^1 = 16$, the estimated conditional probabilities of the answers $y^{1,1} = 15$ and $y^{2,1} = 16$ are $\bar{\omega}^{1,1} = \frac{1}{3}$ and $\bar{\omega}^{2,1} = \frac{2}{3}$ respectively.

Using equations 2 and 3, we can approximate equation 1 as follows:

$$p_{r+1}(\tilde{y}|x) \approx \sum_{m=1}^{M} \sum_{\ell=1}^{L_m} \omega^m \bar{\omega}^{\ell,m} \delta(\tilde{y} - y^{\ell,m}), \tag{4}$$

$$= \sum_{n=1}^{N} \bar{\omega}^n \delta(\tilde{y} - \bar{y}^n). \tag{5}$$

Here, $N$ is the number of distinct answers among $\{y^{\ell,m}\}_{\ell=1,m=1}^{L_m,M}$. The probability of having $\bar{y}^n$ as the answer is estimated as:

$$\bar{\omega}^n = \sum_{m=1}^{M} \sum_{\ell=1}^{L_m} \omega^m \bar{\omega}^{\ell,m} \mathbf{1}(y^{\ell,m} = \bar{y}^n). \tag{6}$$

From Figure 1, we observe that at $r = 2$, we have $N = 4$, the distinct answers are $\bar{y}^1 = 15$, $\bar{y}^2 = 16$, $\bar{y}^3 = 17$, and $\bar{y}^4 = 18$. As shown in eq. 6, the probability of obtaining $\bar{y}^4 = 18$ is $\bar{\omega}^4 = (\frac{2}{4} \times \frac{1}{3}) + (\frac{1}{4} \times 1) = \frac{5}{12}$. We observe that the probability of obtaining the correct answer is increased in one round of HM.

We can stop this procedure by applying a variety of stopping criteria. For example, we can stop (i) after a fixed number of iterations (when $r > R$); or (ii) based on a predefined sampling budget $B_{max}$ (when $r > R$, for $R$ such that $\sum_{p=1}^{R} B_p \leqslant B_{max} < \sum_{p=1}^{R+1} B_p$); or (iii) when the estimate of the mode of $p_r(y|x)$ remains the same for two successive iterations. Algorithm 1 provides a pseudocode description.

---

**Algorithm 1** Hint Marginalization (HM)

---

1: **Input:** task $x$
2: **Hyperparameters:** sampling budget $B_{max} > 0$, number of iterations $R > 1$ and $\{B_r > 0\}_{r=1}^{R}$ such that $\sum_{r=1}^{R} B_r = B_{max}$
3: **Output:** answer $\hat{y}$, approximations of $\{p_r(\tilde{y}|x)\}_{r=1}^{R}$
4: **for** $r = 0 : R - 1$ **do**
5:     **if** $r = 0$ **then**
6:         Sample $B_1$ answers from $p_1(\cdot|x)$.
7:         Approximate $p_1(\tilde{y}|x)$ using eq. 2.
8:     **else**
9:         **for** $m = 1 : M$ **do**
10:           Sample $\lfloor \frac{B_{r+1}}{M} \rfloor$ answers from $p(\cdot|x, \text{Hint}(y^m))$ in order to form a Monte carlo approximation, as shown in eq. 3.
11:         **end for**
12:         Approximate $p_{r+1}(\tilde{y}|x)$ using eqs. 5 and 6.
13:     **end if**
14:     Find the mode of the approximated $p_R(\tilde{y}|x)$ and assign it to $\hat{y}$.
15: **end for**

---

### 3.4 DISCUSSION

Intuitively, some hints are more promising or useful than others for solving a given reasoning task. Our HM framework naturally defines the usefulness of a hint $y'$ at the end of $r$-th iteration by the numerical value of $p_r(y'|x)$, and subsequently weights the answers generated using the hint $y'$ at the $(r+1)$-th iteration by this value, while updating the distribution of answers in eq. 1.

We also note that the HM framework is agnostic to the choice of prompts and is generally applicable to any advanced prompting techniques, since those methods combined with SC can be used for initializing $p_1(\tilde{y}|x)$ for subsequent Hint Marginalization. Our contribution is thus orthogonal to prompt engineering approaches.

## 4 EXPERIMENTAL RESULTS

### 4.1 BENCHMARKS

We evaluate the proposed HM algorithm on six arithmetic benchmark datasets: AddSub (Hosseini et al., 2014), consisting of math word problems requiring addition and/or subtraction for solution; MultiArith (Roy and Roth, 2015), containing math problems requiring multiple reasoning steps for solutions; SingleEQ (Koncel-Kedziorski et al., 2015), consisting of questions which can be solved using a single equation; SVAMP (Patel et al., 2021), containing math word problems of varying structures; GSM8K (Cobbe et al., 2021), consisting of grade school level math word problems; and AQuA (Ling et al., 2017), containing algebraic word problems. In summary, AddSub and SingleEq contain easier problems, whereas the tasks in MultiArith, SVAMP, GSM8K, and AQuA are more challenging since they require multi-step reasoning for a solution. Although these arithmetic problems are relatively simple for humans, LLMs often struggle in solving these types of problems (Patel et al., 2021). All these benchmarks are available under open-source licenses [1].

### 4.2 MODELS

We use three different language models: GPT-3.5 Turbo (Brown et al., 2020), which is fine-tuned using RLHF (GPT-3 based model, unreleased number of parameters), its upgraded version GPT-4 Turbo (OpenAI et al., 2024), and the more recent cost-efficient GPT-4o-mini. All of these models are closed-source, but can be publicly accessed using the OpenAI API at https://openai.com/api.

### 4.3 BASELINES AND EXPERIMENTAL SETTING

We compare our approach to few-shot CoT (Wei et al., 2022b), its combination with SC (Wang et al., 2023), PHP (Zheng et al., 2023), and PHP+SC. We refer to the proposed algorithm as CoT+HM, since the same few-shot prompt as CoT is employed to initialize our approach. For relatively cheaper LLMs, GPT-3.5 Turbo and GPT-4o-mini, we also consider another variant of our method called PHP+HM, where the initial answer distribution is obtained from several PHP provided answers (i.e., PHP+SC). Whenever results are available for other recent iterative refinement methods such as Self-Refine (Madaan et al., 2023), CRITIC (Gou et al., 2023), repeated introspection (Self-Convinced prompting (Zhang et al., 2023a)), Multi-Agent (Debate) (Du et al., 2023), and multi-agent multi-model round table conference (ReConcile (Chih-Yao Chen et al., 2023)), we compare HM with them as well. However, as indicated by Huang et al. (2024) and confirmed by our experimental results, these approaches are significantly outperformed by CoT+SC. So, we do not run these methods for the other datasets and/or LLMs, if the results are not available in the corresponding papers. We conduct our experiments on an Intel(R) Xeon(R) Gold 6140 CPU @ 2.30GHz, and access the GPT models through the OpenAI API.

For a fair comparison with CoT+SC, which requires sampling of multiple CoTs, we ensure that the proposed HM use a comparable number of CoTs. We use a total budget of $B_{max}=40$ sampled CoTs in two iterations of CoT+HM, with $B_1=5$, $B_2=15$, and $B_3=20$. We allocate more CoTs to the later iterations ($r > 1$), since we need to estimate $p(\tilde{y}|x, y')$ for multiple values of the hint $y'$. Since, we initialize $p_1(\tilde{y}|x)$ with CoT+SC, increasing the number of CoTs does not contribute substantially to

---

Table 1: Mean and standard error of accuracy (in %) of few-shot arithmetic reasoning. The **highest** accuracy among all competing algorithms using the same LLM is marked in **bold** and is shown in **red**, **blue**, and **orange** for **GPT-3.5 Turbo**, **GPT-4 Turbo**, and **GPT-4o-mini** respectively. The second-best accuracy in those cases is marked with an underline and is shown in light red, light blue, and light orange respectively. The **highest** accuracy is marked with an asterisk if the difference from the second-best accuracy is statistically significant.

| LLM | Algorithm | AddSub | MultiArith | SingleEQ | SVAMP | GSM8K | AQuA |
|---|---|---|---|---|---|---|---|
| GPT-3.5 Turbo | CoT | 91.4±1.4 | 97.8±0.6 | 97.0±0.7 | 81.9±1.2 | 78.2±1.1 | 58.3±3.1 |
| | PHP | 91.6±1.4* | 99.2±0.4 | 97.6±0.7 | 83.4±1.2 | 83.2±1.0 | 59.1±3.1 |
| | CoT+SC | 91.1±1.4 | 99.0±0.4 | 97.6±0.7 | 85.1±1.1 | 83.2±1.0 | 69.3±2.9 |
| | PHP+SC | 90.6±1.5 | 98.8±0.4 | 97.4±0.7 | 83.3±1.2 | 85.2±1.0 | 64.2±3.0 |
| | Self-Refine | - | - | - | - | 75.1 | - |
| | CRITIC | - | - | - | 83.3 | 78.2 | - |
| | Self-Convinced | 79.3 | - | - | 84.9 | 81.5 | 62.0 |
| | Multi-Agent (Debate) | - | - | - | - | 85.0±3.5 | - |
| | ReConcile | - | - | - | - | 85.3±2.2 | 66.0±0.8 |
| | CoT+HM | 91.6±1.4* | 99.7±0.2* | 98.0±0.6 | 86.2±1.1* | 87.5±0.9 | 70.5±2.9* |
| | PHP+HM | 91.4±1.4 | 99.3±0.3 | 98.4±0.5* | 85.9±1.1 | 88.6±0.9* | 70.5±2.9* |
| GPT-4 Turbo | CoT | 96.5±0.9 | 98.3±0.5 | 96.5±0.8 | 92.3±0.8 | 86.4±0.9 | 83.9±2.3 |
| | PHP | 96.5±0.9 | 98.5±0.5 | 97.4±0.7 | 93.3±0.8 | 91.4±0.8 | 83.9±2.3 |
| | CoT+SC | 96.2±1.0 | 98.8±0.4 | 97.0±0.8 | 93.4±0.8 | 88.5±0.9 | 85.8±2.2* |
| | PHP+SC | 95.9±1.0 | 98.8±0.4 | 96.9±0.8 | 93.9±0.8 | 91.1±0.8 | 82.7±2.3 |
| | CoT+HM | 96.5±0.9 | 98.8±0.4 | 98.6±0.5* | 94.6±0.7* | 94.6±0.6* | 84.3±2.3 |
| GPT-4o-mini | CoT | 92.9±1.3 | 98.8±0.4 | 94.5±1.0 | 93.5±0.8 | 91.5±0.8 | 78.7±2.5 |
| | PHP | 93.9±1.2 | 98.8±0.4 | 95.3±0.9 | 93.6±0.8 | 93.2±0.7 | 78.7±2.6 |
| | CoT+SC | 92.9±1.3 | 98.8±0.4 | 95.1±1.0 | 94.0±0.8 | 93.6±0.7 | 82.7±2.4 |
| | PHP+SC | 92.9±1.3 | 98.8±0.4 | 95.1±1.0 | 93.4±0.8 | 93.4±0.7 | 84.3±2.3 |
| | CoT+HM | 94.4±1.2 | 98.8±0.4 | 95.7±0.9 | 94.1±0.7 | 94.3±0.6* | 84.6±2.3 |
| | PHP+HM | 96.5±0.9* | 98.8±0.4 | 98.4±0.6* | 94.3±0.7* | 94.3±0.6* | 85.0±2.2* |

improved performance at $r = 1$ (Aggarwal et al., 2023). For PHP+HM, we perform one round of hint marginalization with $B_1=20$ and $B_2=20$.

For the CoT+SC algorithm, we sample exactly 40 CoTs (as in (Wang et al., 2023)) to report its performance. On the other hand, generating one answer from PHP requires at least 2 interactions, but the exact number of CoTs cannot be known beforehand. So, in order to ensure a fair comparison, we collect PHP answers in the PHP+SC algorithm until the total number of LLM calls matches that of CoT+HM. This ensures that PHP+SC has an inference time comparable to that of CoT+HM. Except for CoT and PHP, which use greedy decoding, a temperature of 0.7 is used for all sampling based approaches, following the experimental settings of (Wang et al., 2023; Zheng et al., 2023). The answer extraction and cleansing is carried out by following the same steps laid out by Kojima et al. (2022). Additionally, for all datasets except AQuA (where the answers are multiple choice between A-E), we use a 3'rd decimal rounding off of LLM answers and 'ground truth' before comparing them. This fixes some questions in most of those five datasets for all competing algorithms (e.g. the 'true' answer is 0.066666, but the LLM's answer is 0.067), where the LLM's answer is essentially correct, but is declared incorrect due to a rounding error. We measure the accuracy of the answer as the performance metric. CoT employs the same 4-shot prompt for AQuA and the same 8-shot prompt for all other datasets, as designed by Wei et al. (2022b). PHP and PHP+SC also use the same base prompts to obtain the initial answer(s). Example prompts for all algorithms can be found in Section 8.2 of the Appendix.

## 4.4 RESULTS

We summarize the experimental results in Table 1. For each dataset and LLM, we conduct a Wilcoxon signed rank test between the top two algorithms and declare their difference statistically significant at the 5% level. As we use more recent versions of the GPT models than in the original articles of CoT+SC (Wang et al., 2023) and PHP (Zheng et al., 2023), the results are not directly comparable, but are broadly in line with their reported numbers. We observe that for all LLMs, with or without

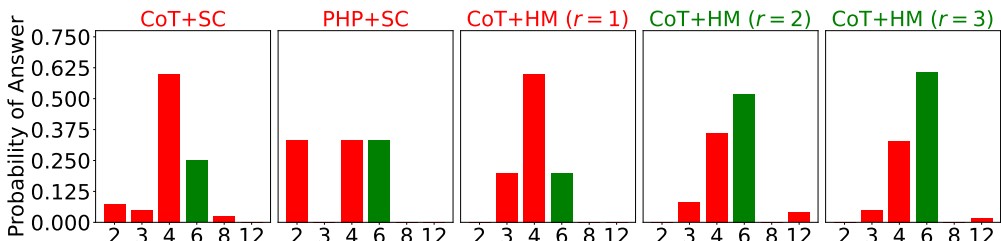

Figure 2: The estimated probabilities of different answers from CoT+SC, PHP+SC, and CoT+HM (using GPT-3.5 Turbo) for an example from GSM8K dataset.
**Question:** The ice cream parlor was offering a deal, buy 2 scoops of ice cream, get 1 scoop free. Each scoop cost \$1.50. If Erin had \$6.00, how many scoops of ice cream should she buy? **Answer: 6**.

SC, PHP achieves higher accuracy than CoT prompting in most cases, demonstrating the advantage of using the LLMs' answers as hints. The superior accuracy of CoT+SC compared to the greedy decoding of CoT for the majority of datasets showcases the strong empirical performance of SC, arising due to the consideration of diverse reasoning paths. PHP+SC emerges as a close competitor to CoT+SC in most cases, although the relative accuracy gain compared to PHP is much lower, since PHP in itself is a strong baseline. Since PHP+SC does not consistently outperform CoT+SC, we can conclude that the incorporation of hints alone is insufficient to achieve better reasoning accuracy.

Our approach, CoT+HM, considerably outperforms CoT+SC in most cases. The PHP+HM variant performs comparably to CoT+HM on GPT-3.5 Turbo but shows improved performance on GPT-4o-mini. This shows that our HM approach is generally applicable, as it can be combined with different prompting methods for initialization, and it is not overly sensitive to the choice of hyperparameters.

One benchmark that deviates from this pattern is AQuA on GPT-4 Turbo, where the best performing procedure is CoT+SC. This might be due to the fact that AQuA is the only multiple-choice question-answering benchmark among the six, and the employed hinting prompt "The answer is close to a)" makes less sense for these types of questions. Further research on how to better extend PHP's hinting prompt to these types of problems might be valuable. In addition, all methods perform only as well as (or even worse than) a vanilla few-shot CoT and PHP on AddSub for both GPT-3.5 Turbo and GPT-4 Turbo models, possibly indicating the fact that the gains to be had using advanced methods on a dataset containing relatively simple questions are rather limited.

Figure 2 shows the estimated probabilities of different answers of an example question from GSM8K for all sampling based algorithms using GPT-3.5 Turbo. We observe that, while both CoT+SC and PHP+SC fail to reason correctly, the proposed CoT+HM outputs the correct answer at both $r=2$ and 3, although its initial distribution (computed using CoT+SC with $B_1=5$ samples) does not have a mode at the correct answer. More interestingly, CoT+SC cannot fix the error even if the budget increases to 40 from 5. On the contrary, the proposed CoT+HM utilizes the additional inference cost effectively to increase the probability of the correct answer at each iteration, demonstrating the usefulness of performing HM in multiple iterations.

While Figure 2 shows that CoT+HM has a higher probability of the correct answer for a specific example question, a dataset-level investigation is necessary to determine whether this phenomenon is general. To that end, we restrict ourselves to only the 'difficult' questions in these benchmarks. If a question is solved correctly by all algorithms in Table 1, we categorize it as 'easy'. A question that is not 'easy' is termed 'difficult'. All easy questions are subsequently removed from the datasets[2]. For all 'difficult' questions, we rank CoT+SC, PHP+SC, and CoT+HM in terms of the probability they assign to the correct answer. The stacked-histograms of these ranks for all six datasets using GPT-4o-mini are shown in Figure 3. We observe that the proposed CoT+HM achieves the lowest rank based on the probability of correct answer across all 'difficult' questions for all datasets more often, outperforming both CoT+SC and PHP+SC. This demonstrates that CoT+HM has higher probability of the correct answer compared to its competitors for most of these 'difficult' questions, which

---

[2]Since in each of these datasets, the majority of the questions are 'easy', all of CoT+SC, PHP+SC, and CoT+HM methods assign a very high probability on the correct answers for them. In order to bring out the differences among these algorithms, we only focus on the 'difficult' questions.

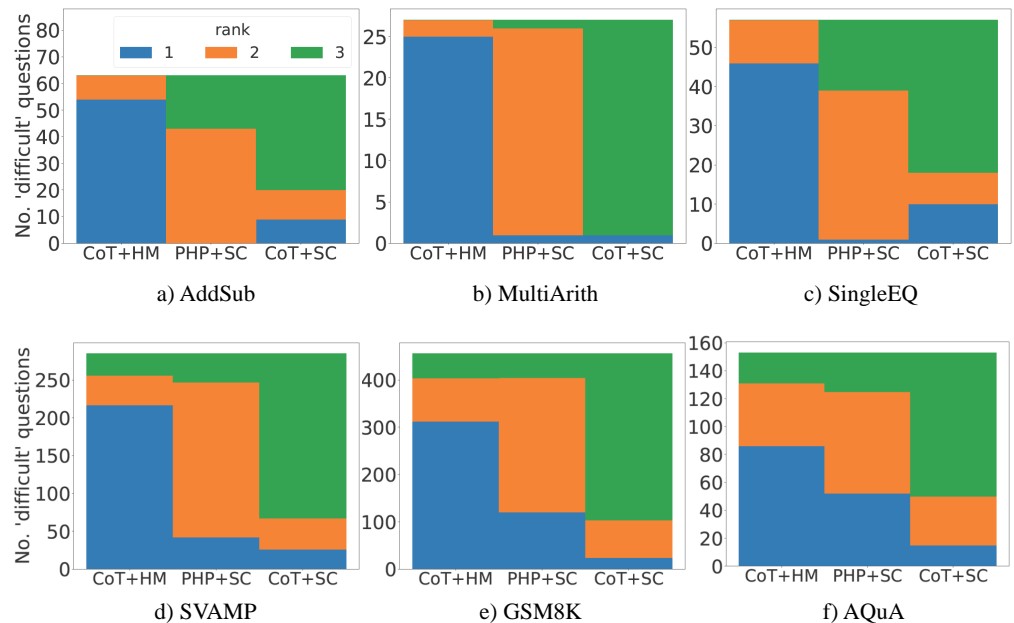

Figure 3: Histogram of **ranks of the algorithms (highest probability of the correct answer results in the lowest rank)** for the **'difficult'** questions from all six datasets using **GPT-4o-mini**

supports our intuition, presented in Section 3.1. Similar results are obtained for the other two LLMs (see Appendix 8.9).

## 5 RELATED WORK

Our proposed method can be situated within a larger literature that aims to improve LLMs' reasoning ability through iterative refinement of chains-of-thought. These works primarily differ in the strategy used to refine the reasoning.

In Progressive Hint Prompting (PHP) (Zheng et al., 2023), chains-of-thought are repeatedly generated, each time providing the previous answers as hints to the LLM. However, it is unlikely that all the hints are equally useful for solving the task, and this approach does not have any mechanism for differential treatment of hints. Two subsequent works propose alternative prompts to steer the answer away from, rather than closer to, the hint. Progressive Rectification Prompting (Wu et al., 2024) proposes a prompt of the form "The answer is likely not <hint>", whereas Deliberate-then-Generate (Li et al., 2023) assumes an error was committed and asks the LLM to identify and correct the mistake. Our method could likely be adapted to those variants by replacing PHP's hinting prompt by their criticism prompt in our procedure. Also closely related to PHP and our work is Hint-before-Solving Prompting (Fu et al., 2024), which triggers the LLM to generate a "hint" before solving a problem. These hints are key ideas that can be used to solve a problem, such as a mathematical formula or a general direction. Unlike PHP or our work, however, there is no iterative aspect where the hints depend on previous attempts at answering the question. Contrary to our work, none of these methods target the refinement of the *distribution* of answers.

Besides these works that aim to iteratively improve answers, there is a growing literature of works that seek to achieve the same goal more explicitly using verbal criticism, at the cost of increased complexity. Self-Refine (Madaan et al., 2023) incorporates a prompt where the LLM self-criticizes its answer, before being queried again with this reflection. Generative Agents (Park et al., 2023) uses a similar procedure, albeit in the context of an agent interacting with an environment. CRITIC (Gou et al., 2023) is a more general framework, where the criticism prompt can make use of external tools like a web search engine to offer grounded corrections. In a different direction, Self-Convinced Prompting (Zhang et al., 2023a) and Reflexion (Shinn et al., 2023) expand on Self-Refine by adding extra modules such as a separate answer encoder, or separating the evaluation and self-reflection dimensions of criticism into separate modules. Other related approaches include multi-

round debate (Du et al., 2023) and consensus via weighted voting mechanism (Chih-Yao Chen et al., 2023). Recent studies have, however, cast doubt on the ability of LLMs to self-criticize effectively (Huang et al., 2024; Tyen et al., 2023), leading researchers to consider using a separately trained LLM as the critic.

REFINER (Paul et al., 2023) fine-tunes a separate critic by supervised learning on examples perturbed by hand-designed rules and GPT-3.5 Turbo. Retroformer (Yao et al., 2023) and RL4F (Akyürek et al., 2023) consider fine-tuning of the critic using reinforcement learning instead, which allows for a more precise alignment with the task of improving answers. As applied to chain-of-thought, these methods have in common that they offer a procedure that takes a chain-of-thought and analyzes it to produce an improved version with a potentially different answer. A crucial difference between these methods and our approach is that these algorithms generate a sequence of chains-of-thought, whereas we propose to refine the *distribution* of answers.

Finally, our work can be seen within the greater context of trying to improve chain-of-thought reasoning within large language models. In existing work, several directions for improving CoTs are considered, including construction of better prompts to aid the LLM in reasoning (Fu et al., 2023; Zhang et al., 2023b), fine-tuning with CoTs (Zelikman et al., 2022) so that the LLMs learn to reason, and effective exploration strategies for multi-hop reasoning (Besta et al., 2023; Yao et al., 2023). A recent survey by Chu et al. (2023) provides a comprehensive overview of these techniques. Our contribution is orthogonal to these prompting techniques since we consider improving the *distribution* of answers iteratively rather than focusing on individual CoTs. Novel variants of HM can be constructed by using these methods for initialization.

## 6 LIMITATIONS

Our Hint Marginalization approach relies on hints having an impact on the answers provided by the LLM. Our focus was not on prompt design, so in our experiments, we employed the hint structure employed by (Zheng et al., 2023): "The answer is close to <hint>". While this hint can be effective for arithmetic reasoning tasks, it is less suitable for other types of answers, such as proper names or options for a multiple choice questions (e.g. AQuA). This motivates research into a more general strategy for effectively structuring the prompt to guide an LLM towards a known or likely answer, while allowing some flexibility to navigate to other answers.

In terms of computation, although the LLM calls in each round of our method can be performed in parallel, the rounds themselves must be performed sequentially. In settings where there is no bound on how many LLM calls can be executed in parallel, CoT+SC can thus make more effective use of the parallelism to reduce latency.

## 7 CONCLUSION

This work presents a novel algorithmic approach, Hint Marginalization, to enable an LLM to solve a reasoning task by iteratively refining its inference distribution. The proposed algorithm addresses the issue of the diminishing marginal utility of extra LLM calls for Self-Consistency. Hint Marginalization focuses on the distribution over the answers at each stage and assigns weights to the hints accordingly, concentrating on promising hints. The marginalization procedure improves sample efficiency. The experimental results, over a range of arithmetic benchmarks and several LLM variants, provide strong evidence that the approach leads to improved reasoning for the same budget of LLM calls, compared to Self-Consistency and other state-of-the-art refinement approaches.

The work can be extended in several directions. Our experiments focus on arithmetic reasoning tasks, and employ a hint in the prompt that is most suitable for such tasks. For other types of problems, alternative hint structures are more natural, and likely to lead to better performance. For example, in tasks that require a verbal response, the prompt could incorporate "verbal criticism", based on one of the approaches detailed in Section 5. In the current version of the procedure, we assign the same number of LLM calls to each unique answer from the previous round. Investigating more efficient strategies to allocate LLM calls non-uniformly to different hints could be another worthwhile direction.

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

# 8 APPENDIX

## 8.1 HINTING PROMPT FROM ZHENG ET AL. (2023)

A typical arithmetic reasoning question is presented in Table 2, where the chain-of-thought yielded the incorrect answer $y = 21$. In Table 3, the PHP hinting prompt is applied to this same question with the hints $y' = 4, 7$, yielding a chain-of-thought with the correct answer $y = 6$.

Table 2: Example problem taken from Zheng et al. (2023).

---

Q: There are 15 trees in the grove. Grove workers will plant trees in the grove today. After they are done, there will be 21 trees. How many trees did the grove workers plant today?

A: There are 21 trees in the Grove today. Grove workers are done planting trees. So 21 trees were planted in the Grove.

---

## 8.2 EXAMPLES OF FULL PROMPTS

We provide examples of full prompts for CoT in Table 4, PHP in Table 5 and our method CoT+HM in Table 6. As CoT+SC and PHP+SC are simply sampled versions of CoT and PHP, the prompts are identical as in those cases. The base examples are taken from Zheng et al. (2023).

Table 3: Demonstration of the hinting prompt from Zheng et al. (2023), as applied to the example problem from Table 2. Additions are highlighted in blue.

---

Q: There are 15 trees in the grove. Grove workers will plant trees in the grove today. After they are done, there will be 21 trees. How many trees did the grove workers plant today? (Hint: The answer is near to 4, 7).

A: We know the Answer Hints: 4, 7. With the Answer Hints: 4, 7, we will answer the question. There are 15 trees originally. Then there were 21 trees after the Grove workers planted some more. So there must have been 21 - 15 = 6 trees that were planted. The answer is 6.

---

Table 4: An example of a CoT prompt with 4 few-shot examples.

---

Q: There are 15 trees in the grove. Grove workers will plant trees in the grove today. After they are done, there will be 21 trees. How many trees did the grove workers plant today?
A: There are 15 trees originally. Then there were 21 trees after the Grove workers planted some more. So there must have been 21 - 15 = 6 trees that were planted. The answer is 6.

Q: If there are 3 cars in the parking lot and 2 more cars arrive, how many cars are in the parking lot?
A: There are originally 3 cars. Then 2 more cars arrive. Now 3 + 2 = 5 cars are in the parking lot. The answer is 5.

Q: Leah had 32 chocolates and her sister had 42. If they ate 35, how many pieces do they have left in total?
A: Originally, Leah had 32 chocolates and her sister had 42. So in total they had 32 + 42 = 74. After eating 35, they had 74 - 35 = 39 pieces left in total. The answer is 39.

Q: Jason had 20 lollipops. He gave Denny some lollipops. Now Jason has 12 lollipops. How many lollipops did Jason give to Denny?
A: Jason had 20 lollipops originally. Then he had 12 after giving some to Denny. So he gave Denny 20 - 12 = 8 lollipops. The answer is 8.

Q: Shawn has five toys. For Christmas, he got two toys each from his mom and dad. How many toys does he have now?
A:

---

## 8.3 VALIDITY OF OUR ASSUMPTIONS

In order to motivate HM in Section 3.1, we considered a hypothetical question $x$, whose answer $\tilde{y} \in \{0, 1\}$ is binary and the true answer is 1. We define $\alpha = p_1(\tilde{y}=1|x)$, and $\gamma = p(\tilde{y}=0|x, \mathrm{Hint}(y=1))$ and $\delta = p(\tilde{y}=1|x, \mathrm{Hint}(y=0))$.

We assumed that $\gamma$ is very small and $\delta > 0$. Both of these assumptions are empirically supported by analysis of obtained results from the PHP (Zheng et al., 2023) technique.

As an example, using GPT-4 Turbo on the entire GSM8K dataset, the empirical frequency of obtaining an incorrect answer conditioned on an immediate correct hint is 0.0179. This suggests that assuming $\gamma$ to be very small is justified. On the other hand, the empirical frequency of obtaining a correct answer conditioned on a previous incorrect hint is 0.3159, which supports the assumption of having a strictly non-zero value for $\delta$. Similar results are obtained for other datasets and LLMs examined in our experiments.

Table 5: An example of a PHP prompt with 4 few-shot examples. The few-shot questions are provided true answer, while the test question is provided with the history so far, in this case $y' = 7, 11, 8$.

---

Q: There are 15 trees in the grove. Grove workers will plant trees in the grove today. After they are done, there will be 21 trees. How many trees did the grove workers plant today? (Hint: The answer is near to 6).
A: We know the Answer Hints: 6. With the Answer Hints: 6, we will answer the question. There are 15 trees originally. Then there were 21 trees after the Grove workers planted some more. So there must have been 21 - 15 = 6 trees that were planted. The answer is 6.

Q: If there are 3 cars in the parking lot and 2 more cars arrive, how many cars are in the parking lot? (Hint: The answer is near to 5).
A: We know the Answer Hints: 5. With the Answer Hints: 5, we will answer the question. There are originally 3 cars. Then 2 more cars arrive. Now 3 + 2 = 5 cars are in the parking lot. The answer is 5.

Q: Leah had 32 chocolates and her sister had 42. If they ate 35, how many pieces do they have left in total? (Hint: The answer is near to 39).
A: We know the Answer Hints: 39. With the Answer Hints: 39, we will answer the question. Originally, Leah had 32 chocolates and her sister had 42. So in total they had 32 + 42 = 74. After eating 35, they had 74 - 35 = 39 pieces left in total. The answer is 39.

Q: Jason had 20 lollipops. He gave Denny some lollipops. Now Jason has 12 lollipops. How many lollipops did Jason give to Denny? (Hint: The answer is near to 8).
A: We know the Answer Hints: 8. With the Answer Hints: 8, we will answer the question. Jason had 20 lollipops originally. Then he had 12 after giving some to Denny. So he gave Denny 20 - 12 = 8 lollipops. The answer is 8.

Q: Shawn has five toys. For Christmas, he got two toys each from his mom and dad. How many toys does he have now? (Hint: The answer is near to 7, 11, 8).
A:

---

## 8.4 EXPERIMENTAL RESULTS USING LLAMA

We have conducted experiments with two Llama-family LLMs: the weaker Llama-3-8b-instruct and the very capable Llama-3-70b-instruct. In order to reduce the API cost of the experiments, we restrict running the more expensive 70B model to only the three most difficult benchmarks.

From the results in Table 7, we observe that using Llama-3-8b-instruct, the relative advantage of PHP over CoT is diminished in comparison to the GPT models. This suggests that weaker LLMs, such as Llama-3-8b-instruct, which often have relatively poor instruction following capability, cannot utilize the hint effectively for solving the reasoning task, highlighting the inadequacy of sophisticated prompting for weaker LLMs. In this setting, the effect of the quality of approximation of the initial distribution of HM becomes important for obtaining a good reasoning accuracy and PHP+HM outperforms CoT+HM in most cases. Except for GSM-8K, PHP+HM either outperforms CoT+SC or obtains comparable performance on all other datasets. On the contrary, for a strongly capable Llama-3-70b-instruct model, both CoT+HM and PHP+HM perform well.

## 8.5 RESULTS ON THE MATH DATASET

In Table 8, we have included results for the MATH dataset, which is a much more challenging mathematical reasoning dataset. For several sub-disciplines (Geometry, Intermediate algebra, Pre-calculus), the *state-of-the-art* performance (without using extreme computation and a very long inference time) is in the range of 50-65 percent, suggesting that LLMs still find these problems very difficult to solve. The proposed HM approach leads to a performance improvement in 5 out of 7 settings.

Table 6: An example of a CoT+HM prompt with 4 few-shot examples. The few-shot questions are provided with true answers as hints, while the test question is provided with one of the distinct answers obtained during the interaction with the LLM in the previous iteration. In this case, $y' = 8$.

---

Q: There are 15 trees in the grove. Grove workers will plant trees in the grove today. After they are done, there will be 21 trees. How many trees did the grove workers plant today? (Hint: The answer is near to 6).
A: We know the Answer Hints: 6. With the Answer Hints: 6, we will answer the question. There are 15 trees originally. Then there were 21 trees after the Grove workers planted some more. So there must have been 21 - 15 = 6 trees that were planted. The answer is 6.

Q: If there are 3 cars in the parking lot and 2 more cars arrive, how many cars are in the parking lot? (Hint: The answer is near to 5).
A: We know the Answer Hints: 5. With the Answer Hints: 5, we will answer the question. There are originally 3 cars. Then 2 more cars arrive. Now 3 + 2 = 5 cars are in the parking lot. The answer is 5.

Q: Leah had 32 chocolates and her sister had 42. If they ate 35, how many pieces do they have left in total? (Hint: The answer is near to 39).
A: We know the Answer Hints: 39. With the Answer Hints: 39, we will answer the question. Originally, Leah had 32 chocolates and her sister had 42. So in total they had 32 + 42 = 74. After eating 35, they had 74 - 35 = 39 pieces left in total. The answer is 39.

Q: Jason had 20 lollipops. He gave Denny some lollipops. Now Jason has 12 lollipops. How many lollipops did Jason give to Denny? (Hint: The answer is near to 8).
A: We know the Answer Hints: 8. With the Answer Hints: 8, we will answer the question. Jason had 20 lollipops originally. Then he had 12 after giving some to Denny. So he gave Denny 20 - 12 = 8 lollipops. The answer is 8.

Q: Shawn has five toys. For Christmas, he got two toys each from his mom and dad. How many toys does he have now? (Hint: The answer is near to 8).
A:

---

Table 7: Mean and standard error of accuracy (in %) of few-shot arithmetic reasoning. The **highest** accuracy among all competing algorithms using the same LLM is marked in **bold** and is shown in **red** and **blue** for **Llama-3-8b-instruct** and **Llama-3-70b-instruct** respectively. The second-best accuracy in those cases is marked with an underline and is shown in light red and light blue respectively.

| LLM | Algorithm | AddSub | MultiArith | SingleEQ | SVAMP | GSM8K | AQuA |
|---|---|---|---|---|---|---|---|
| Llama-3-8b-instruct | CoT | 88.9±1.6 | 96.7±0.7 | 90.0±1.3 | 83.5±1.2 | 76.6±1.2 | 51.2±3.1 |
| | PHP | 90.4±1.5 | 94.7±0.9 | 91.1±1.3 | 86.4±1.1 | 76.8±1.2 | 57.1±3.1 |
| | CoT+SC | 91.1±1.4 | **98.0±0.6** | 94.5±1.0 | **90.4±0.9** | **85.0±1.0** | 59.4±3.1 |
| | CoT+HM | **92.9±1.3** | 96.8±0.7 | 94.9±1.0 | 90.1±0.9 | 82.3±1.1 | 60.0±3.0 |
| | PHP+HM | **92.9±1.3** | 97.8±0.6 | **95.1±1.0** | **90.4±0.9** | 84.2±1.1 | **66.1±3.0** |
| Llama-3-70b-intruct | CoT | - | - | - | 91.2±0.9 | 93.2±0.7 | 72.8±2.8 |
| | PHP | - | - | - | 91.9±0.9 | 93.3±0.7 | 73.2±2.8 |
| | CoT+SC | - | - | - | 92.6±0.8 | 94.2±0.6 | 78.0±2.6 |
| | CoT+HM | - | - | - | **93.1±0.8** | 94.2±0.6 | **79.9±2.5** |
| | PHP+HM | - | - | - | 92.7±0.8 | **94.6±0.6** | 78.7±2.6 |

Table 8: Mean and standard error of accuracy (in %) of reasoning on the Math dataset using GPT-4o-mini. The **highest** accuracy among all competing algorithms is marked in **bold** and the second-best accuracy in those cases is marked with an underline.

| Algorithm | Algebra | Counting and Probability | Geometry | Intermediate Algebra | Number Theory | Prealgebra | Precalculus |
|-----------|---------|-------------------------|----------|---------------------|---------------|------------|-------------|
| **CoT** | 88.5±0.9 | 73.4±2.0 | 55.1±2.3 | 51.5±1.6 | 76.3±1.8 | 86.9±1.1 | 49.1±2.1 |
| **PHP** | 90.2±0.9 | 75.3±2.0 | 55.9±2.3 | 52.3±1.7 | 78.1±1.8 | 87.6±1.1 | 51.1±2.1 |
| **CoT+SC** | 93.9±0.7 | **82.9±1.7** | 64.7±2.2 | 58.1±1.7 | 83.5±1.6 | **91.2±1.0** | 51.3±2.1 |
| **CoT+HM** | 94.1±0.7 | 81.0±1.8 | 64.1±2.2 | 58.3±1.7 | 82.0±1.7 | **91.2±1.0** | 51.5±2.1 |
| **PHP+HM** | **94.8±0.6** | 80.6±1.8 | **65.3±2.2** | **58.9±1.6** | 85.4±1.5 | 90.7±1.0 | **52.0±2.1** |

## 8.6 RESULTS FOR THE 'DIFFICULT' QUESTIONS

In order to demonstrate the advantage of CoT+HM more clearly, we restrict ourselves to only the 'difficult' questions in the six arithmetic benchmarks. If a question is solved correctly by all algorithms in Table 9, we categorize it as 'easy'. A question which is not 'easy' is termed 'difficult'. All easy questions are subsequently removed from the datasets to compute the accuracies only on the difficult questions. From Table 9, we observe that the relative accuracy gains offered by the proposed CoT+HM algorithm are more substantial in most cases.

Table 9: Mean and standard error of accuracy (in %) of few-shot arithmetic reasoning for the 'difficult' questions. The **highest** accuracy among all competing algorithms using the same LLM is marked in **bold** and is shown in **red**, **blue**, and **orange** for **GPT-3.5 Turbo**, **GPT-4 Turbo**, and **GPT-4o-mini** respectively. The second-best accuracy in those cases is marked with an underline and is shown in light red, light blue, and light orange respectively. The **highest** accuracy is marked with an asterisk if the difference from the second-best accuracy is statistically significant.

| LLM | Algorithm | AddSub | MultiArith | SingleEQ | SVAMP | GSM8K | AQuA |
|-----|-----------|--------|------------|----------|-------|-------|------|
| **GPT-3.5 Turbo** | CoT | 46.0±6.3 | 51.9±9.6 | 73.7±5.8 | 36.5±2.9 | 37.1±2.2 | 30.7±3.7 |
| | PHP | 47.6±6.2* | 81.5±7.4 | 78.9±5.4 | 41.8±2.9 | 51.3±2.3 | 32.0±3.7 |
| | CoT+SC | 44.4±6.3 | 77.8±7.9 | 78.9±5.4 | 47.7±3.0 | 51.3±2.3 | 49.0±4.0 |
| | PHP+SC | 41.3±6.3 | 74.1±8.4 | 77.2±5.6 | 41.4±2.9 | 57.2±2.3 | 40.5±4.0 |
| | **CoT+HM** | **47.6±6.3*** | **92.6±5.1*** | **82.5±5.1*** | **51.6±3.0*** | **63.8±2.2*** | **51.0±4.0*** |
| **GPT-4 Turbo** | CoT | 77.8±5.3 | 63.0±9.3 | 68.4±6.2 | 73.0±2.6 | 60.7±2.3 | 73.2±3.6 |
| | PHP | 77.8±5.3 | 66.7±9.2 | 77.2±5.5 | 76.5±2.5 | 75.2±2.0 | 73.2±3.6 |
| | CoT+SC | 76.2±5.4 | **74.1±8.4** | 73.7±5.8 | 76.8±2.5 | 66.7±2.2 | **76.5±3.5*** |
| | PHP+SC | 74.6±5.4 | **74.1±8.5** | 71.9±5.9 | 78.6±2.4 | 74.3±2.1 | 71.2±3.6 |
| | **CoT+HM** | **77.8±5.3** | **74.1±8.4** | **87.7±4.4*** | **81.1±2.3*** | **84.4±1.7*** | 73.9±3.5 |
| **GPT-4o-mini** | CoT | 55.6±6.2 | 74.1±8.5 | 50.9±6.6 | 77.2±2.5 | 75.4±2.0 | 64.7±3.9 |
| | PHP | 61.9±6.2 | 74.1±8.4 | 57.9±6.6 | 77.5±2.5 | 80.3±1.9 | 64.7±3.8 |
| | CoT+SC | 55.6±6.3 | 74.1±8.4 | 56.1±6.6 | 78.9±2.4 | 81.6±1.8 | 71.2±3.7 |
| | PHP+SC | 55.6±6.3 | 74.1±8.5 | 56.1±6.5 | 76.8±2.5 | 80.9±1.9 | 73.9±3.5 |
| | **CoT+HM** | **65.1±6.1*** | 74.1±8.4 | **61.4±6.4*** | **79.3±2.4*** | **83.6±1.7*** | **74.5±3.5*** |

## 8.7 OTHER BIG-BENCH TASKS BEYOND ARITHMETIC REASONING

We provide results for "Date Understanding" and "Object Tracking", which are problems sets involving quantitative (but not strictly mathematical or arithmetic) reasoning.

From the results in Table 10, we observe that PHP still outperforms CoT, demonstrating the utility of hinting beyond the arithmetic tasks. The proposed CoT+HM offers an improvement in accuracy over the baselines for both of these datasets.

Table 10: Mean and standard error of accuracy (in %) of reasoning for Date Understanding and Object Tracking tasks using GPT-4o-mini. The **highest** accuracy among all competing algorithms is marked in **bold** and the second-best accuracy in those cases is marked with an underline.

| Algorithm | Date Understanding | Object Tracking |
|-----------|--------------------|-----------------|
| CoT | 91.9±1.4 | 96.4±0.7 |
| PHP | 93.5±1.3 | 97.7±0.5 |
| CoT+SC | 93.8±1.3 | 96.7±0.7 |
| **CoT+HM** | **94.6±1.2** | **98.0±0.5** |

## 8.8 DISCUSSION ON THE INTUITION OF USING HINTS

As Zheng et al. (2023) note, hinting allows humans to check their answers and improve upon their previous solution to a given problem. We conjecture that in selecting its arithmetic answer, the LLM assigns attention to the hint and in particular, its understanding of the phrase "close to x" provides additional bias towards selecting a number that is closer to the suggested hint.

Additional support for the benefit of hinting is presented by (Fu et al., 2024). In their work, the LLM is encouraged via in-context examples to prepare a hint before solving the problem. The developed hints are more general than those we employ in our work, but the performance improvement in reasoning is indicative of the potential value of a hint in directing an LLM towards a good solution. Further evidence is provided by (Agrawal et al., 2024). In their work, a hint is generated using a weaker LLM. This is observed to yield a performance improvement over multiple math-based reasoning datasets.

## 8.9 ADDITIONAL RESULTS FOR COMPARING PROBABILITY OF CORRECT ANSWER

Figure 3 in the main paper shows that in comparison to CoT+SC and PHP+SC using GPT-4o-mini, CoT+HM assigns higher probability to the correct answers for most of the 'difficult' questions across all datasets. Figures 4 and 5 demonstrate that the same trend holds for both GPT-3.5-Turbo and GPT-4-Turbo LLMs.

Table 11: p-value from Wilcoxon signed rank test between the probabilities of true answers from distributions $p_3(y|x)$ and $p_1(y|x)$ for the 'difficult' questions (for the entire dataset)

| LLM | AddSub | MultiArith | SingleEQ | SVAMP | GSM8K | AQuA |
|-----|--------|-----------|----------|-------|-------|------|
| GPT-3.5 Turbo | 0.0291 (0.1172) | 0.0006 ($1.3\times10^{-5}$) | 0.0012 ($8.6\times10^{-5}$) | 0.0132 ($1.4\times10^{-5}$) | $9.2\times10^{-18}$ ($4.3\times10^{-22}$) | 0.0001 ($1.6\times10^{-8}$) |
| GPT-4 Turbo | 0.2868 (0.2258) | 0.0104 ($2.3\times10^{-6}$) | 0.0002 ($6.2\times10^{-7}$) | $4.8\times10^{-8}$ ($1.7\times10^{-13}$) | $2.2\times10^{-31}$ ($1.5\times10^{-41}$) | 0.0065 (0.0042) |
| GPT-4o-mini | 0.0038 (0.0024) | 0.8413 (0.0243) | 0.0317 (0.0255) | 0.5898 (0.3028) | $4.5\times10^{-12}$ ($5.2\times10^{-12}$) | $2.1\times10^{-5}$ ($8.5\times10^{-6}$) |

Table 12: Percentage of 'difficult' questions (percentage of questions in the entire dataset), so that $p_3(y|x) \geqslant p_1(y|x)$ is satisfied (in other words, HM does not decrease the probability of the true answer)

| LLM | AddSub | MultiArith | SingleEQ | SVAMP | GSM8K | AQuA |
|-----|--------|-----------|----------|-------|-------|------|
| GPT-3.5 Turbo | 79.4 (92.7) | 85.2 (97.3) | 86.0 (97.2) | 63.5 (83.8) | 70.8 (81.4) | 64.7 (74.8) |
| GPT-4 Turbo | 76.2 (95.7) | 96.3 (99.7) | 87.7 (98.0) | 89.5 (96.9) | 85.7 (93.3) | 79.1 (86.6) |
| GPT-4o-mini | 85.7 (97.2) | 96.3 (99.7) | 82.5 (97.0) | 81.1 (93.9) | 83.8 (92.7) | 75.8 (83.9) |

In order to demonstrate the statistical significance of the increase in probability of the true answer, we conduct a Wilcoxon signed rank test between $p_3(y|x)$ (i.e., the estimated probability of the true answer obtained from the proposed CoT+HM) and $p_1(y|x)$ (i.e., the probability of the true answer, at the initialization of CoT+HM, estimated from CoT+SC using 40 samples), and report the p-values in Table 11. We observe that except for 5 out of 36 cases (6 datasets, 3 LLMs, and 2 different partitions of the datasets), the difference between $p_3(y|x)$ and $p_1(y|x)$ is statistically significant at the 5% level, providing strong empirical support in favor of the capability of the HM iterations in increasing the probability of the true answers.

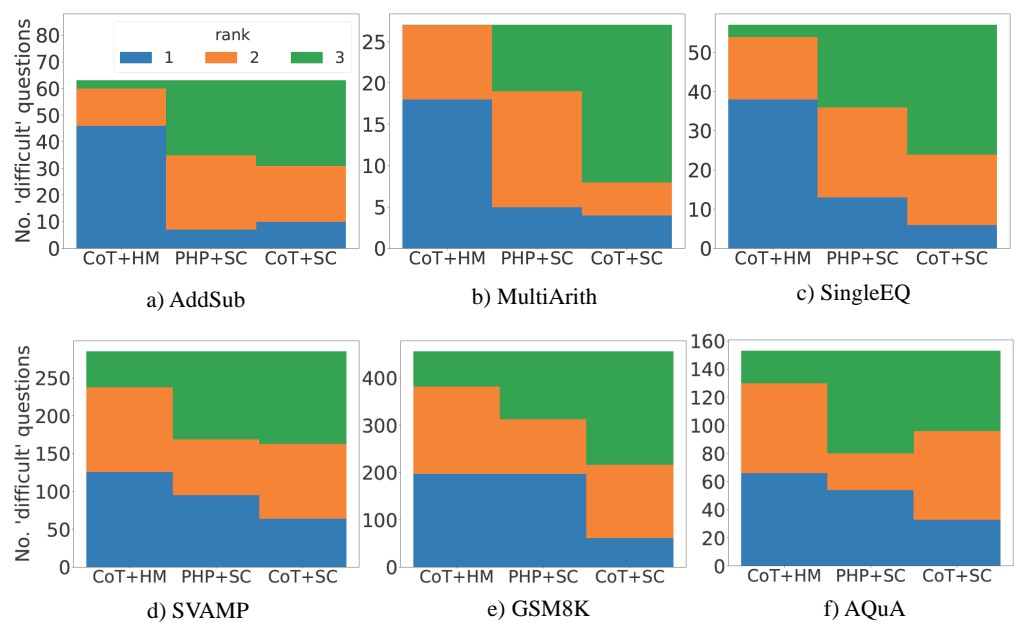

Figure 4: Histogram of **ranks of the algorithms (highest probability of the correct answer results in the lowest rank)** for the **'difficult'** questions from all six datasets using **GPT-3.5 Turbo**

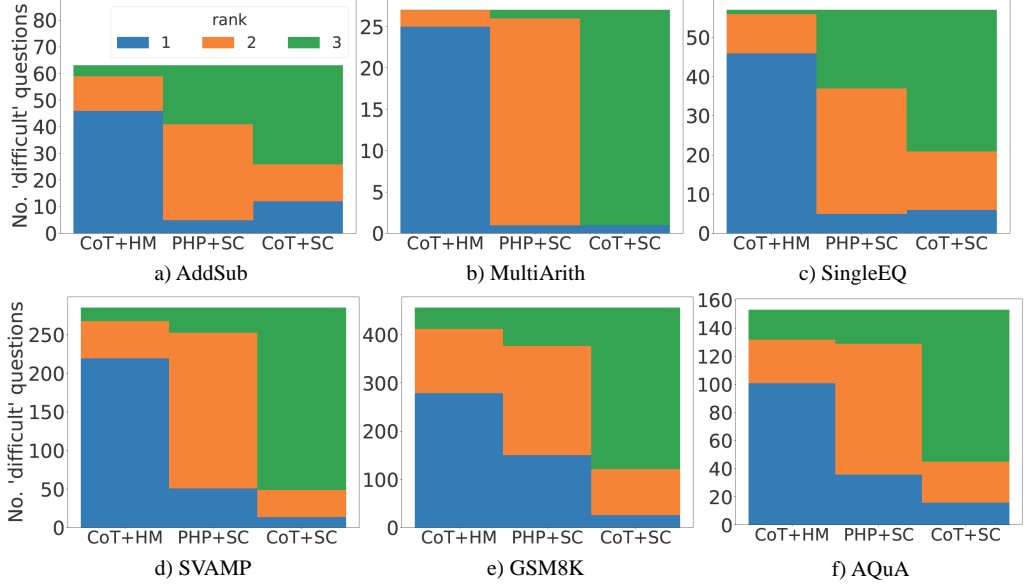

Figure 5: Histogram of **ranks of the algorithms (highest probability of the correct answer results in the lowest rank)** for the **'difficult'** questions from all six datasets using **GPT-4 Turbo**

In addition, we also calculate the percentage of difficult questions for which $p_3(y|x) \geqslant p_1(y|x)$ is satisfied and report the results in Table 12. We observe that in each case, for the majority of the questions, HM iterations do not decrease the probability of the true answer.

