# OpenReview forum: "Hint Marginalization for Improved Reasoning in Large Language Models"
_ICLR.cc/2025/Conference — Submitted to ICLR 2025_

### Official Review · Reviewer_mTdx · 2024-10-27

**Soundness:** 2
**Presentation:** 3
**Contribution:** 1
**Rating:** 5
**Confidence:** 3

**Summary:**

The paper presents Hint Marginalization (HM), an iterative prompting framework for refining for certain kinds of LLM tasks. The paper introduces the method of Hint Marginalization in a general algorithmic way. Intuitively the method can be understood as first sampling an output distribution from multiple attempts of the first prompt, which is then iteratively refined by sampling answer hints from this initial distribution, which are then provided as an additional information in the next round of prompts. This is then supplemented with an experimental evaluation on a number of standard arithmetic reasoning benchmarks for GPT-3.5 Turbo, GPT-4 Turbo and GPT-4o mini.

**Strengths:**

- The empirical evaluation does show some improvements, most importantly in comparision to self-consistency methods.

- The paper proposes a general model for refining LLM answers. Although the type of task seems to implicitly be limited to tasks that have "atomic" answers (e.g., numbers).

- Following the initial distribution of answers in sampling hints is an interesting idea and framing the problem as iteratively refining distributions is a robust foundation for the technique.

**Weaknesses:**

For the presentation as a general method, the experimental evaluation is too narrow. Only arithmetic reasoning benchmarks are tested, which is problematic in two ways.

- For one, within the category of math benchmarks the most challenging benchmarks (such as MATH) are left out. For the chosen benchmarks, state-of-the-art models already perform very well (see e.g., [1]). The reference to Patel et al. 2021 that motivates their choice does not seem timely here either. Employing such extensive prompting regimes for mild improvements in weak models is not a strong motivation. It would be important to see how the method performs also on more challening tasks where there is actually headroom to see improvements over current models.

- Second, a limitation to arithmetic benchmarks seems arbitrary when presenting a general method. Are there limitations that make the application of HM problematic in those other seetings? In particular it seems that the method is limited to reasoning tasks that output a singular (discrete) answer. In more complex settings it is unclear how the hint distribution can be reasonably formed.

- There no experiments with state-of-the-art models such as GPT-4o or any models not by OpenAI. It is unclear to what degree the observed improvements for weaker/old models translate in any meaningful way also to new models or to alternative architectures. Already in the provided experimental data (Table 1) we see that the improvement gains from HM diminish significantly with GPT-4 Turbo. Strikingly, on AQuA self-consistency becomes stronger than HM, wheras the situation was flipped for the weaker GPT-3.5.

[1] Seßler, Kathrin, et al. "Benchmarking Large Language Models for Math Reasoning Tasks." arXiv preprint arXiv:2408.10839 (2024).

**Questions:**

- From the algorithm and the mathemaical exposition it is unclear to me if HM can work for continuous distributions. Could you please elaborate on whether there is any consideration for this setting?

- Is there any intution for why hints regarding proximity to other numbers are helpful to the LLM in arithmetic tasks? For example referring to the first prompt example in 8.2, why would knowledge of the answer being near 4, 7 be helpful in reasoning? There is no causal way to arrive at the answer from this knowledge and the output of the prompt does not seem to take this into account. I understand that the hints condition the LLM, but it is unclear to me why such conditioning would be helpful in general.

---

> ### Author Response · Authors · 2024-11-24
> **Response to Reviewer mTdx**
>
> We thank the reviewer for acknowledging the generality and 'robust foundation' of our work. Below, we address your concerns.
>
> > #  W1. a) For the presentation as a general method, the experimental evaluation is too narrow. Only arithmetic reasoning benchmarks are tested, which is problematic in two ways. For one, within the category of math benchmarks the most challenging benchmarks (such as MATH) are left out.
>
> The lack of particularly challenging datasets is a valid criticism of our work, also raised by other reviewers. We have now included results for the MATH dataset (please refer to Table 8 of our revised paper), which is a much more challenging mathematical reasoning dataset. For several sub-disciplines (Geometry, Intermediate algebra, Pre-calculus), the state-of-the-art performance (without using extreme computation and a very long inference time) is in the range of 50-65 percent, suggesting that LLMs still find these problems very difficult to solve.  The proposed HM approach leads to a performance improvement in 5 out of 7 settings.
>
> > # W1. b) For the chosen benchmarks, state-of-the-art models already perform very well (see e.g., [1]). The reference to Patel et al. 2021 that motivates their choice does not seem timely here either. Employing such extensive prompting regimes for mild improvements in weak models is not a strong motivation. [1] Seßler, Kathrin, et al. "Benchmarking Large Language Models for Math Reasoning Tasks." *arXiv preprint arXiv:2408.10839* (2024).
>
> We agree that some of the arithmetic  datasets are relatively easy for GPT, but we still observe consistent improvement over self-consistency in most cases.  The same benchmarks are considered in the papers proposing the relevant baselines. Our proposed hint marginalization strategy solves problems where self-consistency fails to establish the correct mode (please refer to Figure 2) and sampling more CoTs is not helpful. If we restrict ourselves to the 'difficult' questions during the performance assessment (eliminating the easy questions that are answered correctly by all LLMs and all methods; please refer to Table 10 of our revised paper), then the improvement is more substantial.
>
> > #  W1. c) It would be important to see how the method performs also on more challenging tasks where there is actually headroom to see improvements over current models.
>
> We agree with the reviewer that applying the method more broadly to other diverse and challenging reasoning domains is a worthwhile and very interesting research direction. With a view to partially satisfying this request, we now provide results for "Date Understanding'' and "Object Tracking'', which are problems sets involving quantitative (but not strictly mathematical or arithmetic) reasoning.
>
> We observe an improvement over the baselines for both of these tasks. The baseline performance is still relatively strong for these datasets, but we note that hint marginalization reduces the average error rate  by more than 10 percent for both tasks compared to the next best baseline. The datasets are often padded with many very easy questions that are answered without difficulty by all methods and LLMs. The performance on the more challenging subset of questions, where some (or all) LLMs make errors is more interesting to analyze. As highlighted by Figure 3 in the paper, our proposed algorithm achieves more noticeable benefits on these subsets of challenging questions. This is also true for the Date Understanding and Object Tracking datasets.
>
> > # W2. a) Second, a limitation to arithmetic benchmarks seems arbitrary when presenting a general method. Are there limitations that make the application of HM problematic in those other settings?
>
> This is a valid point. The suggested inclusion of the Math dataset, as well as the analysis of the Date Understanding and Object Tracking datasets, addresses this limitation.
> Our results on the Math dataset (please refer to Table 8 of our revised paper) and the big-bench reasoning tasks such as Date Understanding and Object Tracking (please refer to Table 9 of our revised paper) demonstrates that our frameworks is advantageous beyond arithmetic reasoning. The Math dataset probes the capabilities of the approach for more general mathematical reasoning (geometry and algebra, for example), and the date understanding and object tracking probe other types of numerical reasoning.
>
> In terms of limitations, extending beyond quantitative reasoning problems for a generalization of the HM framework to other reasoning domains would require careful prompt engineering for designing effective hinting strategies for other domains. If there is not a quantitative answer, there is also a challenge of how to define a distribution and aggregate over different responses. This direction is very interesting but is beyond the scope of our current work.

---

> ### Author Response · Authors · 2024-11-24
> **Cont'd**
>
> > # W2. b) In particular it seems that the method is limited to reasoning tasks that output a singular (discrete) answer. In more complex settings it is unclear how the hint distribution can be reasonably formed.
>
> Our approach is designed for reasoning tasks where there is one *"correct"* answer and the metric is task accuracy (whether the algorithm's answer matches the *"correct"* answer). We note that this problem setting does encompass a wide range of reasoning tasks across various domains, e.g., arithmetic (correct answer is a number), mathematical (correct answer is, for example, an algebraic expression), logical (correct answer is a boolean variable), and multiple-choice questions (with a predefined number of options). The reviewer is correct in observing that this is a limitation of our method, but it is a relatively broad limitation, and still leaves our approach applicable to many reasoning tasks.
>
> We agree that HM (as well as any baseline algorithms considered in this work) is not suitable in its current form for nuanced open-ended question answering. (Please refer to the detailed response below for the discussion of *"continuous distributions"*).
>
> > # W3. a) There no experiments with state-of-the-art models such as GPT-4o or any models not by OpenAI.
>
> Unfortunately, GPT-4o is prohibitively expensive (USD 10.00 / 1M output tokens). We consider that it is sufficient to conduct experiments with multiple LLMs. Aside from this, the documented performance of GPT-4o is not significantly better than GPT-4 Turbo or GPT-4o-mini. We agree that it is important to extend analysis beyond the GPT family, and we now include results for two Llama models (please refer to Table 7 in our revised paper).
>
> > #  W3.  b) It is unclear to what degree the observed improvements for weaker/old models translate in any meaningful way also to new models or to alternative architectures.
>
> For the benchmark experiments in our paper, we use GPT-4o-mini, which was released on July 18, 2024, and is OpenAI's *"most cost-efficient small model that’s smarter and cheaper than GPT-3.5 Turbo"* ([source](https://openai.com/api/pricing/)).
> This model was thus released very recently.  Based on the performance of GPT-4o-mini in Table 1 and its release date, it cannot be viewed as one of the *"weaker/old models"*. Its performance is close to that of GPT-4o.
>
> However, we acknowledge that it is important to conduct experiments with other architectures. Hence, we now include experimental results for two Llama-3 variants (please refer to Table 7 in our revised paper).
>
> > # W3. c) Already in the provided experimental data (Table 1) we see that the improvement gains from HM diminish significantly with GPT-4 Turbo. Strikingly, on AQuA self-consistency becomes stronger than HM, whereas the situation was flipped for the weaker GPT-3.5.
>
> We agree with the reviewer that some of the arithmetic datasets are relatively easy for GPT-4-Turbo, but we still observe that CoT+HM provides accuracy improvement over self-consistency in 15 out of 18 cases in Table 1, which strongly supports the general usefulness of the proposed HM approach.
>
> Moreover, the new experiments on Math (please refer to Table 8 of our revised paper) and other big-bench tasks (please refer to Table 9 of our revised paper) show the general usefulness of HM beyond these benchmarks.

---

> ### Author Response · Authors · 2024-11-24
> **Cont'd**
>
> > # Q1. a) From the algorithm and the mathematical exposition it is unclear to me if HM can work for continuous distributions.
>
> Our approach is designed for reasoning tasks where there is one  'correct' answer and the metric is task accuracy (whether the algorithm's answer matches the `correct' answer). The most relevant baseline algorithms we consider, e.g. CoT+SC and PHP, are also suitable only in the same setting.
>
> Chen et al., 2023 points out that self-consistency can only be applied to tasks where the final answer is a number, a True/False boolean variable, or an option (a)/(b)/(c) from a multiple-choice set. Without considerable modification, self-consistency cannot handle tasks that involve free-form generation, such as code generation, translation, summarization, and open-ended, descriptive question answering. **Thus, it is not a limitation of the proposed Hint Marginalization arising from our design, but is common to all relevant baselines.**
>
> However for those tasks with a single 'correct' answer, we (and self-consistency as well) do not make any explicit assumptions regarding the distribution of answers (i.e., discrete or continuous) and do not assume any a priori knowledge of the support.
> Our iterative sampling and marginalization procedure provides a valid Monte Carlo approximation of the sequence of distributions of answers, defined in Eq. 1 of the paper irrespective of whether $p_r(y|x)$ is continuous or discrete.
>
> Note that, depending on the nature of the target answer and the evaluation protocol in a task, one can introduce further approximations for applying HM. For example, if for a question, we have the prior knowledge that the answer is an integer, and the LLM outputs a float, we could use a round-off after each round of HM. If the evaluation protocol only requires a match of the algorithm's answer to the 'correct' answer up to two decimal points, we should perform a two decimal points round off for all answers and hints for all LLM sampled answers. Alternatively, one could instruct the LLM explicitly to provide integer answers/ answers up to two decimal points. If the answer is an option between 'yes/no', then a careful answer extraction and parsing would allow us to group different versions of the same answer (e.g. 'yes', 'Yes', 'YEAH', 'Certainly' etc.) into the same category and sum their probabilities. We could also ask the LLM via another call to group all answers in the two distinct categories after each round.
>
> This aspect is **not unique to our HM approach; for evaluation of relevant baselines such as self-consistency, the same consideration is required**. Often, the dataset is grouped together with the code for answer parsing and evaluation, provided by the dataset curators to ensure a fair evaluation.
>
> Chen, Xinyun, et al., "Universal Self-Consistency for Large Language Model Generation" arxiv preprint arXiv:2311.17311 (2023).
>
> > # Q1. b) Could you please elaborate on whether there is any consideration for this setting?
>
> As discussed above, it is not entirely clear whether the reviewer refers to free-form language generation tasks as 'continuous distribution'. If this is the case, our method will require adjustment, similar to the modifications proposed by Chen et al., 2023 in adapting self-consistency to such tasks. For example, one could use a similar prompt to their 'Universal Self Consistency prompt' to score different generations, and use those scores to form the conditional probabilities $p(\tilde{y}|x, \textit{Hint}(y'))$.
>
> If the reviewer is instead referring to tasks where the answer is real-valued (and hence there is a continuous distribution over candidate answers), then our method does work as is in such settings (Please see the discussion above).

---

> ### Author Response · Authors · 2024-11-24
> **Cont'd**
>
> > # Q2. a) Is there any intuition for why hints regarding proximity to other numbers are helpful to the LLM in arithmetic tasks? For example referring to the first prompt example in 8.2, why would knowledge of the answer being near 4, 7 be helpful in reasoning? There is no causal way to arrive at the answer from this knowledge and the output of the prompt does not seem to take this into account.
>
> First, we would like to note that we do not propose the hint prompt in this work. Rather, it has been adapted to the HM framework from PHP (Zheng et al., 2023) and we do not claim any optimality of its design.
>
> However, since we do make use of the hint mechanism, we can provide some clarification of the intuition behind the mechanism.
> As Zheng et al. (2023) note, hinting allows humans to check their answers and improve upon their previous solution to a given problem.  We conjecture that in selecting its arithmetic answer, the LLM will assign attention to the hint and, in particular, its understanding of the phrase *"close to x"* will provide additional bias towards selecting a number that is closer to the suggested hint. In this way, presence of the hint in the prompt nudges the LLM to consider the hint both as it selects the steps in the rationale and when it answers the question.
>
> Empirically, we observe that there is a significantly greater chance of selecting the same answer as the provided hint. For example, as specified in Appendix 8.3, for the GSM8K dataset, the probability of obtaining an incorrect answer conditioned on providing a correct hint is 0.0179. By contrast, we see that the best performing procedure has an error rate of 0.054. This provides evidence that the insertion of the hint is affecting the answer (in a positive way), even if it is not immediately discernible (as the reviewer correctly point out "There is no causal way to arrive at the answer from this knowledge and the output of the prompt does not seem to take this into account.") in the formation of the rationale for the in-context examples (e.g. in Table 3 in Appendix 8.2).
>
> We argue that investigation of how the LLM is using the hints requires the development of a deeper theoretical understanding of LLMs' few-shot learning capabilities. This is an open question in LLM research at present, but is not the main contribution or focus of this paper.
>
> Additional support for the benefit of hinting is presented by Fu et al. (2024). In their work, the LLM is encouraged via in-context examples to prepare a hint before solving the problem. The developed hints are more general than those we employ in our work, but the performance improvement in reasoning is indicative of the potential value of a hint in directing an LLM towards a good solution. Further evidence is provided by Agrawal et al. (2024). In their work, a hint is generated using a weaker LLM. This is observed to yield a performance improvement over multiple math reasoning datasets.
>
> We understand that the paper did not sufficiently explain the intuition and value of hinting and we have modified the paper to include a summary of this discussion in Appendix 8.8, citing these two recent works in support.
>
> ### References:
>
> - Fu, Jinlan, et al. "Hint-before-Solving Prompting: Guiding LLMs to Effectively Utilize Encoded Knowledge." *arXiv preprint arXiv:2402.14310* (2024).
> - Agrawal, Vansh, et al. "Give me a hint: Can LLMs take a hint to solve math problems?" *arXiv preprint arXiv:2410.05915* (2024).
>
> > # Q2. b) I understand that the hints condition the LLM, but it is unclear to me why such conditioning would be helpful in general.
>
> Please refer to the discussion above for the intuition of using hints.  Empirically, our results in Table 1 (or the results in Table 2 in PHP (Zheng et al., 2024)) show that PHP consistently outperforms CoT, which provides evidence in support of the usefulness of hinting.
>
> Our analysis in Appendix 8.3 shows that:
> - (a) using the correct answer as a hint, the LLM generates the same answer with a very high probability; and
> - (b) even with an incorrect hint in the prompt, the LLMs are at least somewhat likely to generate the correct answer in the next interaction.
>
> Moreover, from Table 9 of our revised paper, we observe that PHP still outperforms CoT in other Big-Bench tasks such as *"Date Understanding"* and *"Object Tracking"*, demonstrating its utility beyond the arithmetic tasks.

---

> ### Author Response · Authors · 2024-11-27
> **Request for Feedback on Rebuttal Responses**
>
> Dear Reviewer mTdx,
>
> As the discussion period nears its end, we hope that we have effectively addressed and resolved your concerns.
>
> Your feedback on our rebuttal responses would be greatly appreciated. We are more than happy to provide further clarification on any remaining issues.
>
> Thank you for your time and consideration.

---

> ### Author Response · Authors · 2024-12-01
> **Response to Reviewer mTdx**
>
> We thank the reviewer for reading our rebuttal.
>
> > # Experimental performance
>
> For completeness, we copy Table 8 from the paper to make discussion easier.
>
> Mean and standard error of accuracy (in \%) of reasoning on the Math dataset using GPT-4o-mini. The **highest** accuracy among all competing algorithms is marked in **bold** and the _second-best_ accuracy in those cases is marked in _italic_.
>
> | **Algorithm**     | **Algebra**  | **Counting and Probability** | **Geometry**  | **Intermediate Algebra** | **Number Theory** | **Prealgebra** | **Precalculus** |
> |-------------------|--------------|------------------------------|---------------|--------------------------|-------------------|----------------|-----------------|
> | **CoT**           | 88.5±0.9     | 73.4±2.0                     | 55.1±2.3      | 51.5±1.6                 | 76.3±1.8          | 86.9±1.1       | 49.1±2.1        |
> | **PHP**           | 90.2±0.9     | 75.3±2.0                     | 55.9±2.3      | 52.3±1.7                 | 78.1±1.8          | 87.6±1.1       | 51.1±2.1        |
> | **CoT+SC**        | 93.9±0.7     | **82.9±1.7**                 | *64.7±2.2*    | 58.1±1.7                 | *83.5±1.6*        | **91.2±1.0**   | 51.3±2.1        |
> | **CoT+HM**        | *94.1±0.7*   | *81.0±1.8*                   | 64.1±2.2      | *58.3±1.7*               | 82.0±1.7          | **91.2±1.0**   | *51.5±2.1*      |
> | **PHP+HM**        | **94.8±0.6** | 80.6±1.8                     | **65.3±2.2**  | **58.9±1.6**             | **85.4±1.5**      | *90.7±1.0*     | **52.0±2.1**    |
>
> We apologize that our phrasing in the response and the revised paper was unclear. We intended to refer to the performance of PHP+HM, not the grouped performance of the HM-based techniques. As can be seen in the table above, **the proposed PHP+HM does obtain the best accuracy in 5 out of 7 sub-categories.**
>
> Regarding the reported results  for Llama-3-70b-instruct, over the three more challenging arithmetic reasoning datasets, **CoT+HM achieves a performance improvement** over the best baseline in 2 out of 3 cases (equal in the third case), with an **average performance improvement of 0.8\%**. PHP+HM **outperforms in all three cases**, with an **average accuracy improvement of 0.4\%**.
>
> The **original primary criticism of the review** was that **"the experimental evaluation is too narrow"**.  In response to this, **we included results for Math (identified by the reviewer as a more challenging dataset), for two open models (Llama variants), and two non-arithmetic tasks (Date Understanding and Object Tracking)**.
>
> Now **it appears that the main criticism has changed from the experimental evaluation being too narrow to the observed performance improvement not being large enough**.
>
> The experiments now **encompass 5 LLMs and 9 datasets**. Taking into account the 7 different subcategories of questions in the Math dataset, **we investigate 36 experimental scenarios**. Of these, the **proposed PHP-HM method outperforms all baselines in 26 cases**.  Compared to the **best baseline method**, there is **almost no additional computational overhead introduced by the proposed method**. Although the improvements are not dramatic, they are **observed consistently across multiple datasets and LLMs**.  Using the **Math dataset** as a **challenging example recommended by the reviewer**, the proposed method either **(i) achieves a >0.5\% improvement in 5 out of 7 subcategories for almost no additional computation; or (ii) achieves a 3-10\% improvement compared to less computationally demanding baselines.**
>
> Given that **the paper introduces a novel, principled method**, we consider that **this level of relatively consistent outperformance is more than satisfactory for a research paper**. While we respect the reviewer's opinion, **there seems to be too much focus on the sole criterion of "does the proposed method improve by more than $x$ percent."**
>
> > # Concerns about hinting
>
> Our arguments in favor of using hinting (and in particular the utilization of the PHP (Zheng et al., 2023)-style prompt) in our proposed HM framework can be summarized as follows:
>
> - In Section 3.1, **we show mathematically**  that in the proposed HM framework, if the **'in-flow'** of probability to the 'correct' answer **exceeds** the **'out-flow'** of probability from the 'correct' answer, then the **probability of the correct answer increases** with each HM iteration. Note that this implication goes both ways (**'if and only if'**).
>
> - Thus, if there is any refinement strategy, which **satisfies this 'in-flow' vs 'out-flow' criterion**, then it **becomes a suitable candidate** to be incorporated in the **proposed HM framework**.
>
> - We conduct **detailed analysis of the obtained results** (illustrations in **Figures 3-5**, empirical results in **Tables 11-12**) to demonstrate that there is **strong empirical evidence** that **hinting satisfies this criterion**.
>
> **Continued in the next Official Comment**

---

> ### Author Response · Authors · 2024-12-01
> **Cont'd**
>
> - Figures 3-5 show that the proposed CoT+HM (uses $p_3(\cdot|x)$ for inference) **achieves the lowest rank
> based on the probability of correct answer across the 'difficult' questions for all datasets and all LLMs more often**, outperforming both CoT+SC (uses $p_1(\cdot|x)$ for inference) and PHP+SC.
>
> - The **height of the blue bar** in the **CoT+HM column**, which **counts how many times CoT+HM has higher probability of the correct answer compared to either of the other two algorithms**, is **the largest for all datasets and LLMs** in **Figures 3-5**.
>
> - Remember that for each question, obtaining a **higher probability on the right answer** from (one or more rounds of) CoT+HM than that of CoT+SC is possible **if and only if** the **"in-flow of probability to the correct answer is greater than out-flow of probability from the correct answer''**.
>
> - In order to demonstrate the **statistical significance** of our result, we have now conducted a **Wilcoxon signed rank test** between $p_3(y|x)$ (i.e., the estimated probability of the 'correct' answer obtained from the proposed CoT+HM) and $p_1(y|x)$ (i.e., the probability of the 'correct' answer, at the initialization of CoT+HM, estimated from CoT+SC using 40 samples), and report the **p-values in Table 11 in Appendix 8.9** in the revised version of the paper (**also shown here**).
>
> **p-value from Wilcoxon signed rank test between the probabilities of true answers from distributions $p_3(y|x)$ and $p_1(y|x)$ for the 'difficult' questions (for the entire dataset)**
>
> | **LLM**              | **AddSub**                 | **MultiArith**            | **SingleEQ**             | **SVAMP**                | **GSM8K**                | **AQuA**                 |
> |----------------------|----------------------------|---------------------------|--------------------------|--------------------------|--------------------------|--------------------------|
> | **GPT-3.5 Turbo**    | 0.0291 (0.1172)            | 0.0006 ($1.3 \times 10^{-5}$) | 0.0012 ($8.6 \times 10^{-5}$) | 0.0132 ($1.4 \times 10^{-5}$) | $9.2 \times 10^{-18}$ ($4.3 \times 10^{-22}$) | 0.0001 ($1.6 \times 10^{-8}$) |
> | **GPT-4 Turbo**      | 0.2868 (0.2258)            | 0.0104 ($2.3 \times 10^{-6}$) | 0.0002 ($6.2 \times 10^{-7}$) | $4.8 \times 10^{-8}$ ($1.7 \times 10^{-13}$) | $2.2 \times 10^{-31}$ ($1.5 \times 10^{-41}$) | 0.0065 (0.0042) |
> | **GPT-4o-mini**      | 0.0038 (0.0024)            | 0.8413 (0.0243)            | 0.0317 (0.0255)           | 0.5898 (0.3028)          | $4.5 \times 10^{-12}$ ($5.2 \times 10^{-12}$) | $2.1 \times 10^{-5}$ ($8.5 \times 10^{-6}$) |
>
> - We observe that except for 5 out of 36 cases (6 datasets, 3 LLMs, and 2 different partitions of the datasets), the **difference between $p_3(y|x)$ and $p_1(y|x)$ is statistically significant at the 5\% level**, providing **strong empirical support** in favor of the **capability of the HM iterations in increasing the probability of the 'correct' answers**.
>
> - In addition, we also **calculate the percentage of difficult questions for which $p_3(y|x) \geqslant p_1(y|x)$ is satisfied** and report the results in **Table 12 in Appendix 8.9** in the revised version of the paper (**also shown here**).
>
> **Percentage of 'difficult' questions (percentage of questions in the entire dataset), so that $p_3(y|x) \geqslant p_1(y|x)$ is satisfied (in other words, HM does not decrease the probability of the true answer)**
>
> | **LLM**              | **AddSub**       | **MultiArith**    | **SingleEQ**      | **SVAMP**         | **GSM8K**         | **AQuA**          |
> |----------------------|------------------|-------------------|-------------------|-------------------|-------------------|-------------------|
> | **GPT-3.5 Turbo**    | 79.4 (92.7)      | 85.2 (97.3)       | 86.0 (97.2)       | 63.5 (83.8)       | 70.8 (81.4)       | 64.7 (74.8)       |
> | **GPT-4 Turbo**      | 76.2 (95.7)      | 96.3 (99.7)       | 87.7 (98.0)       | 89.5 (96.9)       | 85.7 (93.3)       | 79.1 (86.6)       |
> | **GPT-4o-mini**      | 85.7 (97.2)      | 96.3 (99.7)       | 82.5 (97.0)       | 81.1 (93.9)       | 83.8 (92.7)       | 75.8 (83.9)       |
>
> - We observe that **in each case**, for the **majority of the questions, HM iterations do not decrease the probability of the 'correct' answer**.
>
> In summary, these results provide **strong and direct empirical evidence** that **hinting** is indeed an **effective strategy for refinement of the answer distribution**, as proposed in our HM framework.

---

### Official Review · Reviewer_5edW · 2024-11-03

**Soundness:** 3
**Presentation:** 3
**Contribution:** 3
**Rating:** 6
**Confidence:** 4

**Summary:**

Hinting has proved itself as a viable approach to improve the reasoning capabilities of an LLM. A common approach is to incorporate a potential answer into the prompt, e.g. by adding "the solution might be close to X" at the end. The submission proposes a simple yet principled approach to leverage the initial answers of an LLM as hints, defining an iterative refinement of the answer distribution. More precisely, given an initial query $q$, one defines probability distributions $p_n(x\mid q)$ recursively by
\begin{align*}
p_0(x\mid q) &= p_{\text{LLM}}(x\mid q) \\\\
p_{n+1}(x\mid q) &= \int p_n(x_0\mid c)p_{\text{LLM}}(x\mid q, \mathrm{HINT}(x_0)])d x_0
\end{align*}
The main motivation behind this definition is the empirical observation that in many cases across the possible hints the flow of probability to the correct answer will be larger than the one incorrect one. As an intuition, one can state that that giving a correct answer as hint will often make the correct answer much more likely while an incorrect answer as hint will often be ignored by the model.

The authors conduct extensive experiments both with multiple datasets and multiple SOTA LLMs and compare hint marginalization against other reasoning frameworks such as self consistency, chain of thought, and progressive hint prompting. Throughout the experiments, hint marginalization consistently shows that it is able to outperform previous methods.

Overall, I think this is a good submission. The proposed idea might be simple but it is also sound and effective. Hence, I can recommend the paper for acceptance.
*Update*: After discussion I decided to downgrade the rating due to the rather weak motivation of the method.

**Strengths:**

- Independent of underlying task.
- Can be combined with other advanced prompting strategies.
- Defines a sound stochastic process as basis for combining answers from multiple hints.
- Provides a simple sampling based algorithm to estimate marginal probabilities iteratively.
- Takes great care to ensure fair evaluation in the experiments.
- Impressive experimental results.

**Weaknesses:**

- Justification of the methods stems only from intuition and a few empirical evidences.

**Questions:**

- Is there any clustering of equivalent answers (e.g. 0.5, 1/2, \frac{1}/{2}, ...) during sampling?
- In the methodology section you define the output of the LLM to consist of the answer $y$ and an additional rationale $z$. Is $z$ used in any way in your procedure?

---

> ### Author Response · Authors · 2024-11-24
> **Response to Reviewer 5edW**
>
> We thank the reviewer for acknowledging the merit of our work. Below, we address your concerns.
>
> > # W1. Justification of the methods stems only from intuition and a few empirical evidences.
>
> In addition to presenting the main experimental results on standard benchmarks in Table 1, we present a simple intuition behind hint marginalization, other quantitative analyses, illustrations, and a case study to highlight the comparison between the proposed HM and relevant sampling-based baseline algorithms.
>
> The assumptions in Section 3.1 are formed by analyzing the PHP results.  An example for GSM8K is provided in Appendix 8.3, which confirms that:
> - (a) using the correct answer as the hint, the LLM generates the same answer with a very high probability; and
> - (b) even with an incorrect hint in the prompt, the LLMs are at least somewhat likely to generate the correct answer in the next interaction.
>
> Similar results are obtained for other datasets and LLMs examined in our experiments.
> We will provide all such results in a table in the revised version and add a reference to Appendix 8.3 in Section 3.1 for a clearer presentation.
>
> We illustrate the steps of Algorithm 1 in Figure 1 for an easier exposition of HM via an example. In addition, Figure 2 shows a case study of how HM corrects an erroneous answer of CoT+SC by increasing the probability of the *"correct"* answer iteratively.
>
> Figures 3-5 show that CoT+HM has a higher probability of the correct answer compared to its competitors (including CoT+SC) for most of the *"difficult"* questions across all datasets and LLMs used in our experiments.  Since CoT+HM is initialized with CoT+SC, these results provide direct empirical evidence that marginalizing over hints indeed increases the probability of the *"correct"* answer.  This justifies our intuitions and demonstrates the efficacy of the proposed HM framework beyond improved accuracies.
>
> >  # Q1. Is there any clustering of equivalent answers (e.g. 0.5, 1/2, $\frac{1}{2}$, ...) during sampling?
>
> The answer extraction and cleansing of answers from sampled CoTs for all algorithms is carried out by following the same steps laid out by Kojima et al. (2022). This involves careful regular expression based parsing of the CoTs and subsequent conversion of each sampled answer from string to float format with (possible) round-off. This allows us to sum the probabilities of the same answer expressed in different formats (as in the example provided by the reviewer), and reduces the number of LLM calls in subsequent iterations of hint marginalization.
>
> > #  Q2. In the methodology section you define the output of the LLM to consist of the answer $y$ and an additional rationale $z$. Is $z$ used in any way in your procedure?
>
> We do not use $z$ explicitly in our procedure and this is a very interesting suggestion and worth further study. Currently we group responses purely based on the final answer, and we form hints without using $z$. There is likely to be valuable information in the produced $z$ that can be exploited (this is supported by the recent reported success of  **process-based** training as opposed to **outcome-based**).
>
> Previous experimental work has suggested that it is important to encourage the LLM to generate $z$, because otherwise we often do not observe a diversity of reasoning with multiple different candidate answers.
> Wang et al. (2023) show that sampling multiple answers for a question and performing a majority vote improves performance only if the LLM is encouraged to generate diverse reasoning paths (e.g., by using few-shot CoT prompting (Wei et al., 2022)) .
>
> We could explicitly use the rationales if we had access to a verifier that is capable of scoring the 'correctness' of the rationales.
> For example, the verification scores of the rationales corresponding to the mode of the distribution of the answers after each round could be utilized to design a stopping criterion for HM. This would be advantageous in allocating computational budget dynamically across tasks with varying difficulty levels.

---

> ### Author Response · Authors · 2024-11-27
> **Thanks for reading the rebuttal**
>
> Thank you very much for acknowledging our rebuttal and appreciating our work. Your positive viewpoint on our work is truly important and encouraging to us.

---

### Official Review · Reviewer_bgqX · 2024-11-04

**Soundness:** 2
**Presentation:** 3
**Contribution:** 2
**Rating:** 6
**Confidence:** 3

**Summary:**

This paper proposes a new protocol for iteratively prompting an LLM to solve reasoning problems.

The protocol is based on the hypothesis that, when the prompt to the LLM contains a candidate answer as a "hint" (e.g., "The answer is probably close to $y$"), the LLM will: (1) output $y$ with high probability if it is in fact the correct answer, and (2) still have some chance of outputting the correct answer even if the hint $y$ is not correct. If this hypothesis holds, then we can iteratively concentrate probability on the correct answer by repeatedly prompting the LLM with its previous answer as a "hint." More formally, we can estimate a sequence of distributions $p_r(y \mid x) = \sum_{y' \in \mathcal{Y}} p_{r-1}(y' \mid x) \cdot p_{LM}(y \mid \text{question}=x, \text{hint}=y')$, where each $p_r$ concentrates more probability on the correct answer than $p_{r-1}$. The paper's proposed method is a particular Monte Carlo scheme for estimating $p_r$, which works by iteratively estimating $p_0, p_1, p_2,$ and so on, with a different sampling budget at each step. The steps of the algorithm must be performed in sequence, but within each step, approximation can be done in parallel.

The paper compares the proposed method to other prompting protocols on six benchmarks with three OpenAI LLMs, and shows that it achieves slightly higher accuracy overall.

**Strengths:**

1. The paper is clearly written and was easy to follow.

2. The paper clearly states the reasons that the technique might be expected to work.

3. The empirical evidence does seem to show that the proposed technique delivers some performance gains on well-studied benchmarks.

**Weaknesses:**

1. I don't really work in this area, and am unfamiliar with ICLR's norms for papers that essentially present a new prompting technique with benchmark results. But I believe the paper does not currently contribute a significant advance to scientific knowledge in this area. For example:

- After reading the paper, I still have very little intuition for why this sort of "hinting" ("The answer is close to X") is supposed to be beneficial. In the example few-shot prompts, the rationales do not appear to use the hint at all. In practice, do LLM rationales use the hints? How? There is no real empirical analysis beyond the overall "our strategy does better" benchmark results.

- The theory of the paper is based on the availability of some iterative-refinement strategy that has high "in-flow" of probability to the correct answer and low "out-flow" of probability away from the correct answer. But there is no systematic study of what sorts of iterative refinement strategies have this property. For example, do the various critique-based prompting strategies satisfy this property? Does "hinting" as you do actually satisfy this property on a wide range of examples? Why?

2. I don't have a sense of the difficulty of the tasks for GPT-4-caliber language models--GPT-4 seems to score in the mid-to-high 90s. What sorts of problems does this technique really help to solve? On harder reasoning benchmarks where GPT-4 still does very poorly (e.g., Chollet's Abstraction and Reasoning Challenge) does this technique actually help? If not, how do the authors view the limitations of this technique?

**Questions:**

1. Some results are starred in your table but I could not find any description of what stars indicate. Can you clarify?

2. What exactly is PHP+HM? I didn't see PHP clearly described, and it was unclear to me how you were using PHP within HM.

---

> ### Author Response · Authors · 2024-11-24
> **Response to Reviewer bgqX**
>
> We thank the reviewer for acknowledging that the paper is 'clearly written' and 'easy to follow'. Below, we address your concerns.
>
> > # W1. a) I don't really work in this area, and am unfamiliar with ICLR's norms for papers that essentially present a new prompting technique with benchmark results.
>
> We would like to stress that our paper **does not** fall into the category of 'papers that essentially present a new prompting technique'.
> In this work, we **do not propose any novel prompt engineering** technique.
> Instead, the HM framework is a **novel iterative hint-based refinement strategy for reasoning with LLMs**, where the main novelty lies in its **capability of maintaining and updating a distribution over answers** for improved reasoning.
>
> In **Section 3.4** of the paper, we discuss that applicability of the proposed HM algorithm is **agnostic to the choice of prompts** and HM can **readily incorporate any advanced prompting techniques**, since those methods combined with SC can be used for
> initializing $p_1(\tilde{y}|x)$ for subsequent iterations of Hint Marginalization.
> **Our contribution is thus orthogonal to prompting approaches.**
>
> Our paper perhaps did not make it sufficiently clear, but in light of the reviewer's comments, we will clarify that **HM is not a prompt design method** in the introduction of the paper.
>
> > # W1. b) But I believe the paper does not currently contribute a significant advance to scientific knowledge in this area.
>
>
> Our contribution (written in detail towards the end of the Introduction Section of the paper) is to propose a novel, probabilistic, simple, computationally efficient, principled, generally applicable, and effective 'iterative refinement strategy' for LLM's reasoning.
>
> **HM** is a **novel** and **probabilistic** framework, since to the best of our knowledge, this is the first work which considers sequential refinement of the **distribution** of LLMs' answers instead of refining **one** answer.
>
> **HM** is remarkably **simple** to implement since the Monte Carlo approximations required for updating the distribution of answers (Eqs. 3 and 6 in the paper) involve straightforward arithmetic calculations only.
>
> **HM** is **computationally efficient** since the runtime of one HM iteration is essentially the same as one LLM call. Implementing Eqs. 3 and 6 contributes negligibly to the runtime, and the LLM calls within each HM iteration using different hints can be carried out in parallel, so that one round of refinement has close to the same latency as that of a single LLM call.
>
> **HM** is **principled** since it formalizes how marginalizing over hints iteratively should make the mode of the inference distribution more likely to be the correct answer under some mild assumptions.
>
> **HM** is **generally applicable** since it is agnostic to the choice of prompts and can readily incorporate any advanced prompting techniques. We experimentally demonstrate how other prompting techniques such as PHP can be combined successfully with our method.
>
> **HM** is **effective** since the results in Table 1 in the paper show that out of 18 experimental scenarios (3 LLMs, six datasets), we observe a statistically significant increase in accuracy in 14. Justifying our intuition, further analyses in Figures 3-5 show that **CoT+HM has a higher probability of the correct answer compared to its competitors more often** across multiple datasets and LLMs.
>
> Therefore, we believe that this work is an **important first step towards developing and understanding better probabilistic inference techniques for LLMs** in the era of OpenAI o1 (which is closed source but is presumed to sample multiple responses and refine them for its final answer to encourage 'slow thinking').

---

> ### Author Response · Authors · 2024-11-24
> **Cont'd**
>
> > # W1. c) For example: After reading the paper, I still have very little intuition for why this sort of "hinting" ("The answer is close to X") is supposed to be beneficial.
>
> We thank the reviewer for drawing our attention to this point.
>
> First, we would like to note that we do not propose the hint prompt in this work. Rather, it has been adapted to the HM framework from PHP (Zheng et al., 2023).
>
> However, since we do make use of the hint mechanism, we can provide some clarification of the intuition behind the mechanism. As Zheng et al. (2023) note, hinting allows humans to check their answers and improve upon their previous solution to a given problem.
>
> We conjecture that in selecting its arithmetic answer, the LLM will assign attention to the hint and in particular, its understanding of the phrase *"close to x"* will provide additional bias towards selecting a number that is closer to the suggested hint.
>
> Additional support for the benefit of hinting is presented by Fu et al. (2024). In their work, the LLM is encouraged via in-context examples to prepare a hint before solving the problem. The developed hints are more general than those we employ in our work, but the performance improvement in reasoning is indicative of the potential value of a hint in directing an LLM towards a good solution. Further evidence is provided by Agrawal et al. (2024). In their work, a hint is generated using a weaker LLM. This is observed to yield a performance improvement over multiple maths reasoning datasets.
>
> We understand that the paper did not sufficiently explain the intuition and value of hinting and we have modified the paper to include a summary of this discussion in Appendix 8.8, citing these two recent works in support.
>
> ### References:
>
> - Fu, Jinlan, et al. "Hint-before-Solving Prompting: Guiding LLMs to Effectively Utilize Encoded Knowledge." *arXiv preprint arXiv:2402.14310* (2024).
> - Agrawal, Vansh, et al. "Give me a hint: Can LLMs take a hint to solve math problems?" *arXiv preprint arXiv:2410.05915* (2024).
>
> > #  W1. d) In the example few-shot prompts, the rationales do not appear to use the hint at all. In practice, do LLM rationales use the hints? How?
>
> We reiterate that we do not propose the hinting prompt in this work and do not claim any optimality of its design.
> Having said that, as the reviewer notes correctly, the rationales do not appear to use the hint explicitly in the example few-shot prompts.
>
> Despite this, the improved accuracy of PHP in comparison to CoT (see Table 1 of our paper and/or Table 2 in the PHP paper) provides strong empirical evidence in support of the usefulness of hint-prompting.
>
> We argue that investigation of how the LLM is using the hints requires the development of a deeper theoretical understanding of LLMs' few-shot learning capabilities. This is an open question in LLM research at present, but is not the main contribution or focus of this paper.
>
> However, we conjecture that the presence of the hint in the prompt nudges the LLM to consider the hint both as it selects the steps in the rationale and when it answers the question. Empirically, we observe that there is a significantly greater chance of selecting the same answer as the provided hint. For example, as specified in Appendix 8.3, for the GSM8K dataset, the probability of obtaining an incorrect answer conditioned on providing a correct hint is 0.0179. By contrast, we see that the best performing procedure has an error rate of 0.054. This provides evidence that the insertion of the hint is affecting the answer (in a positive way), even if it is not immediately discernible in the formation of the rationale.

---

> ### Author Response · Authors · 2024-11-24
> **Cont'd**
>
> > # W1. e) There is no real empirical analysis beyond the overall ``our strategy does better" benchmark results.
>
> We strongly disagree with the assessment that we do not include any empirical analysis beyond the overall *"our strategy does better"*.
>
> In addition to presenting the main experimental results on standard benchmarks in Table 1, we indeed present a simple intuition behind hint marginalization, other quantitative analyses, illustrations, and a case study to highlight the comparison between the proposed HM and relevant sampling-based baseline algorithms.
>
> The assumptions in Section 3.1 are formed by analyzing the PHP results. An example for GSM8K is provided in Appendix 8.3, which confirms that:
> - (a) using the correct answer as the hint, the LLM generates the same answer with a very high probability; and
> - (b) even with an incorrect hint in the prompt, the LLMs are at least somewhat likely to generate the correct answer in the next interaction.
>
> Similar results are obtained for other datasets and LLMs examined in our experiments. We will provide all such results in a table in the revised version and add a reference to Appendix 8.3 in Section 3.1 for a clearer presentation.
>
> We illustrate the steps of Algorithm 1 in Figure 1 for an easier exposition of HM via an example. In addition, Figure 2 shows a case study of how HM corrects an erroneous answer of CoT+SC by increasing the probability of the *"correct"* answer iteratively.
>
> Figures 3-5 show that CoT+HM has a higher probability of the correct answer compared to its competitors (including CoT+SC) for most of the *"difficult"* questions across all datasets and LLMs used in our experiments. Since CoT+HM is initialized with CoT+SC, these results provide direct empirical evidence that marginalizing over hints indeed increases the probability of the *"correct"* answer. This justifies our intuitions and demonstrates the efficacy of the proposed HM framework beyond improved accuracies.
>
> > # W2. a) The theory of the paper is based on the availability of some iterative-refinement strategy that has high "in-flow" of probability to the correct answer and low "out-flow" of probability away from the correct answer. But there is no systematic study of what sorts of iterative refinement strategies have this property.
>
> Our objective in this work is to present a general method and demonstrate its effectiveness via specific instantiations of the framework.
>
> The motivation for choosing the hinting prompt for sequential refinement of the answer distribution stems from:
> - (a) the simplicity; and
> - (b) the effectiveness of the PHP style prompting.
>
> Analyzing whether other iterative refinement strategies have high *"in-flow"* of probability to the correct answer and low *"out-flow"* of probability away from the correct answer is certainly very interesting and would lead to further generalizations of our approach.
>
> However, investigating that aspect is not the main focus of this work for the following reasons:
> 1. Incorporation of other refinement strategies in our framework would require careful prompt engineering and falls outside the scope of this work.
>    As a concrete example, consider self-refine (Madaan et al., 2023), which uses extensive prompt engineering via Python codes and explicitly introduces some errors and corrected versions in the feedback prompt.
>    Incorporating the LLM-generated code as a hint in such a setting would require extensive experimentation with potentially different prompt engineering techniques. This would lead to a very high experimental computational cost because of the complicated nature of such prompts and the need to generate multiple samples for estimating the conditional probabilities.
>
> 2. Our experimental results in Table 1 already show that the proposed HM approaches outperform the iterative refinement methods using the hint-based methodology. We do not believe it is essential to find other refinement approaches that satisfy the *"in-flow"* and *"out-flow"* assumption or to characterize the types of refinement strategies that achieve this.

---

> ### Author Response · Authors · 2024-11-24
> **Cont'd**
>
> > # W2. b) For example, do the various critique-based prompting strategies satisfy this property? Does "hinting" as you do actually satisfy this property on a wide range of examples? Why?
>
> As discussed above, investigation of whether critique-based prompting strategies satisfy the assumptions in Section 3.1 falls outside the scope of this work.  Our analyses (Figures 2-6, Appendix 8.3) provide strong empirical evidence that hinting satisfies the desired properties for the LLMs and datasets considered in our work.
>
> With the additional datasets that are now included in our experimental results, following valuable suggestions by the reviewers, we show that the hint-based approach offers benefits for:
> - Arithmetic reasoning,
> - More general mathematical reasoning including geometry and algebra (MATH dataset),
> - Date understanding and object tracking (Big Bench).
>
> We consider that this demonstrates the applicability of *"hinting"* to a broad range of reasoning tasks.
> We do concur with the reviewer that the investigation of alternative refinement approaches is highly desirable and an exciting avenue to explore.
>
> > #  W3. I don't have a sense of the difficulty of the tasks for GPT-4-caliber language models--GPT-4 seems to score in the mid-to-high 90s. What sorts of problems does this technique really help to solve? On harder reasoning benchmarks where GPT-4 still does very poorly (e.g., Chollet's Abstraction and Reasoning Challenge) does this technique actually help? If not, how do the authors view the limitations of this technique?
>
> We thank the reviewer for this question.  We agree that some of the datasets are relatively easy, but we still observe consistent improvement over self-consistency (a strong baseline) in most cases. The same benchmarks are considered in the papers that have presented the relevant baselines.
>
> Our proposed hint marginalization strategy solves problems where self-consistency fails to establish the correct mode (please refer to Figure 2) and sampling more CoTs is not helpful. If we restrict ourselves to the *"difficult"* questions during the performance assessment (eliminating the easy questions that are answered correctly by all LLMs and all methods; please refer to Table 10), then the improvement is more substantial.
>
> The lack of particularly challenging datasets is a valid criticism of our work, also raised by other reviewers. We have now included results for the MATH dataset (please refer to Table 8 of our revised paper), which is a much more challenging mathematical reasoning dataset. For several sub-disciplines (Geometry, Intermediate algebra, Pre-calculus), the state-of-the-art performance (without using extreme computation and a very long inference time) is in the range of 50-65 percent, suggesting that LLMs still find these problems very difficult to solve. The proposed HM approach leads to a performance improvement in 5 out of 7 settings.
>
> We agree with the reviewer that applying the method more broadly to other reasoning domains is a worthwhile and very interesting research direction.  With a view to satisfying this request, we now provide results for *"Date Understanding"* and *"Object Tracking"*, which are problem sets involving quantitative (but not strictly mathematical or arithmetic) reasoning.
>
> Extending beyond this (outside quantitative problems) would require careful prompt engineering to generalize to other reasoning domains. This direction is very interesting but is not the main focus of our current work.
> Note that the ARC dataset is not amenable to CoT-style prompting and none of the baseline algorithms considered in our work is capable of addressing such problems in their current form.
>
> > # Q1. Some results are starred in your table but I could not find any description of what stars indicate. Can you clarify?
>
> We thank the reviewer for bringing this to our attention. We apologize for not explaining the asterisks in Section 4.4.
> For each dataset and each LLM, we conduct a Wilcoxon
> signed rank test between the top two algorithms and mark the best result with *, if the difference is statistically significant at the 5\% level. Although we mentioned the Wilcoxon test in Section 4.4, we did not explain the asterisk. We have now edited the caption of Table 1 to clarify the use of asterisks.

---

> ### Author Response · Authors · 2024-11-24
> **Cont'd**
>
> > # Q2. What exactly is PHP+HM? I didn't see PHP clearly described, and it was unclear to me how you were using PHP within HM.
>
> As explained in Section 4.3,
> PHP+HM refers to a variant of our method, where the initial answer distribution is approximated using several PHP-provided answers (i.e., PHP+SC).
> In other words, the only difference between CoT+HM and PHP+HM algorithms is in their initializations (using CoT+SC for CoT+HM and PHP+SC for PHP+HM).
> The subsequent HM iterations are carried out in exactly the same manner for both of these algorithms.
>
> In Section 5, we discuss PHP (Zheng et al., 2023), which refers to a sequential refinement based prompting method, where the answers generated by the LLM in previous rounds are used as hints in the prompt to assist the LLM in reasoning.
> The algorithm is terminated when the same answer is repeated in two consecutive rounds. Although PHP uses a hint strategy, it focuses on sequentially refining the prompt. The key difference in our approach is that we refine the distribution over the responses and use hints to guide this refinement.
>
> PHP+SC refers to running several non-interacting PHP chains independently, collecting the terminal answers from each of them, and conducting a majority vote among those answers to determine the solution.

---

> ### Author Response · Authors · 2024-11-27
> **Response to the follow-up questions**
>
> We thank the reviewer for reading our rebuttal and for raising interesting follow up questions. Below, we address your concerns.
>
> >  # I have taken a look at the referenced papers on hinting (Fu et al. and Agrawal et al.). Perhaps I am misreading but they seem to propose a rather different type of "hinting" than in your work. Crucially, their hinting protocols are not techniques for conditioning a language model on a previously generated answer. As such, I do not believe they are relevant to your paper, because those hinting strategies could not be used to refine a distribution into a more concentrated distribution.
>
> We agree that those techniques cannot directly be used to refine a distribution
> into a more concentrated distribution. As written in our rebuttal, our intention was to show support for the general idea of hinting aiding in LLMs' reasoning, not to suggest that these were alternative refinement strategies in the HM framework.
>
> > # As for Zheng et al. 2023, it appears their paper was rejected from TMLR, with multiple reviewers raising as a concern the lack of motivation or ablation testing to understand whether "hinting" really has the claimed properties, e.g.:
>
> > # "Although PHP has improved model performance, its results do not explicitly present any reasoning paths to understand how it utilize these hints."
>
> > # "The prompt looks odd! I don't have much a priori intuition that prompting with not necessarily correct hints would improve the results so significantly, which is why I suspect a lurking alternate explanation."
>
> Although it is true that their paper was rejected from **TMLR**, the authors did submit a version to the **AI4MATH** workshop at **ICML 2024**, which was accepted.  You can read the peer review [here](https://openreview.net/forum?id=UkFEs3ciz8).
>
> Reading the reviews from **TMLR** (which can be read [here](https://openreview.net/forum?id=5HsBuYYx4i)), one gathers that none of the reviewers dispute the fact that the suggested prompt works better than **CoT**.   On the other hand, they were unhappy that the authors had not provided more investigations into why or how the **LLMs** benefited from this strategy.
>
> Your concern falls under the same category. While we consider that a thorough investigation to explain exactly how the **LLM** is impacted by the hint is beyond the scope of our work (and would essentially constitute another paper), we believe that we have provided a reasonable conjecture. More importantly, in alignment with your suggestion, what is critical for our purposes is demonstrating that the proposed hint satisfies the **in-flow** versus **out-flow** assumptions that support our proposed method.
>
> Below, we provide a detailed response to support the use of the **PHP** prompt in our work.
>
> > #  To reiterate my key concern, I understand the logic of your paper to be:
>
> > # (1) Suppose we have a way to condition a language model on its previous answer, in such a way that in-flow of probability to the correct answer is greater than out-flow of probability from the correct answer.
>
>
> > # (2) Then we can iteratively refine the LM's answer distribution to concentrate more and more mass on the correct answer.
>
>
> > # To my knowledge, no one has convincingly demonstrated point (1) in the literature, so it would fall to your paper to defend the existence of such a prompting method.
>
> We thank the reviewer for bringing up this point, which allows us to discuss this issue in detail.
> We agree that no one has convincingly demonstrated point (1) in the literature, so we do indeed need to demonstrate via empirical analysis that PHP prompting (Zheng et al., 2023) satisfies this property for justifying its use in the proposed HM framework.
>
> Perhaps we did not sufficiently emphasize this aspect in our presentation or in the response, but we did conduct a thorough empirical investigation of this issue in our initial submission (see lines 419-428 in our paper and Figures 3-5) and our results clearly support that using PHP-style prompting indeed satisfies this property.
> In light of the reviewer's comment, we will stress this point in the introduction as a valuable contribution of our work, and we will amend the discussion of Figures 3-5 to emphasize how these results support the claim.
>
> Below, we briefly recap our mathematical framework, explain our experimental procedure for this investigation (also written in lines 418-427 in Section 4.4 of our paper), and reiterate and explain our key results for enhanced clarity.
>
> **Continued in the next Official Comment**

---

> ### Author Response · Authors · 2024-11-27
> **Cont'd**
>
> **Mathematical Framework:**
>
> Let us reuse the notations from the main paper, so $x$ denotes a question and $y$ is its 'correct' answer.
>
> Below, we rewrite the definitions (as written in **lines 139-140 in Section 3.1 of our paper**) of the in-flow of probability to the correct answer and the out-flow of probability from the correct answer for one round of hinting for completeness:
> \begin{align}
> \textrm{in-flow-prob.} (x, y)  &= \sum_{y' \neq y} p_{1}(\tilde{y}{=}y'|x) p(\tilde{y}{=}y|x, \mathrm{Hint}(y'))\,  \tag{1}
> \end{align}
> \begin{align}
> \textrm{out-flow-prob.}(x, y) &= p_{1}(\tilde{y}{=}y|x) \sum_{y' \neq y}  p(\tilde{y}{=}y'|x, \mathrm{Hint}(y))\,  \tag{2}
> \end{align}
> \begin{align}
> \textrm{in-flow-prob.} (x, y)&> \textrm{out-flow-prob.} (x, y)
> \end{align}
>
> $\implies$
> \begin{align}
> \sum_{y' \neq y} p_{1}(\tilde{y}{=}y'|x) p(\tilde{y}{=}y|x, \mathrm{Hint}(y')) > p_{1}(\tilde{y}{=}y|x) \sum_{y' \neq y}  p(\tilde{y}{=}y'|x, \mathrm{Hint}(y)) \tag{3}
> \end{align}
>
> In other words, $\textrm{in-flow-prob.} (x, y)$ denotes the joint probability of the event that for a question $x$, the initial answer was incorrect and after one round of hinting, it was corrected.
> Similarly, $\textrm{out-flow-prob.} (x, y)$ is the joint probability of the event that the initial answer was correct and after one round of hinting, it switched to an incorrect answer.
>
> In the HM framework, we compute
> \begin{align}
> p_{2}(\tilde{y}{=}y|x) = p_{1}(\tilde{y}{=}y|x) p(\tilde{y}{=}y|x, \mathrm{Hint}(y)) + \sum_{y' \neq y} p_{1}(\tilde{y}{=}y'|x) p(\tilde{y}{=}y|x, \mathrm{Hint}(y'))\.  \tag{4}
> \end{align}
> Note that the second term on the right hand side is the in-flow probability to the correct answer via hinting.
>
> One can also write:
> \begin{align}
> p_{1}(\tilde{y}{=}y|x) &= p_{1}(\tilde{y}{=}y|x) \times 1\,
> \end{align}
> \begin{align}
> &= p_{1}(\tilde{y}{=}y|x) \times \bigg[ p(\tilde{y}{=}y|x, \mathrm{Hint}(y)) + \sum_{y' \neq y} p(\tilde{y}{=}y'|x, \mathrm{Hint}(y))\bigg]\,
> \end{align}
> \begin{align}
> &=p_{1}(\tilde{y}{=}y|x) p(\tilde{y}{=}y|x, \mathrm{Hint}(y)) + \sum_{y' \neq y} p_{1}(\tilde{y}{=}y|x) p(\tilde{y}{=}y'|x, \mathrm{Hint}(y))\, \tag{5}
> \end{align}
> Note that the second term is the out-flow probability from the correct answer via hinting.
> Combining these two equations (4) and (5), we see that we also have:
> \begin{align}
> p_{2}(\tilde{y}{=}y|x)>p_{1}(\tilde{y}{=}y|x) \implies \textrm{in-flow-prob.} (x, y)&> \textrm{out-flow-prob.} (x, y)\,.
> \end{align}
> Thus, the implication goes both ways, and $p_{2}(\tilde{y}{=}y|x) > p_{1}(\tilde{y}{=}y|x)$ is satisfied **if and only if** the in-flow probability to the correct answer exceeds the out-flow probability from the correct answer.
> In retrospect, we did not stress the **'only if'** part of this condition.
>
> Note that this is true for subsequent rounds of hinting as well.  In other words,  $p_{3}(\tilde{y}{=}y|x) > p_{2}(\tilde{y}{=}y|x)$, **if and only if** the condition in eq. (3) is satisfied with all $p_1(\cdot|x)$-s replaced by $p_2(\cdot|x)$. In our experiments, $p_{1}(\tilde{y}|x)$ is estimated by sampling multiple CoTs in parallel without any hinting.
> Thus, CoT+SC declares the estimated  mode of $p_{1}(\tilde{y}|x)$ as the final answer.
> The proposed CoT+HM algorithm is initialized with the same $p_{1}(\tilde{y}|x)$.
>
> **Experimental Procedure:**
>
> In each of the six arithmetic datasets, the majority of the questions are ‘easy’ and all of CoT+SC, PHP+SC, and
> CoT+HM methods assign a very high probability on the correct answers for them.
> In order to bring out the
> differences among these algorithms, we only focus on the ‘difficult’ questions. We define 'easy' and ‘difficult’ questions in these benchmarks as follows.
> If a question is solved correctly by all algorithms in Table 1, we categorize it as ‘easy’.
> A question that
> is not ‘easy’ is termed ‘difficult’.
> Thus, the accuracies of all algorithms are 100\% on the `easy' questions and removing them from the dataset does not affect the ranking of different algorithms (in Table 1 and Table 9, different algorithms have the same rankings in terms of their accuracies).
>
> For each of the ‘difficult’ questions, we independently rank CoT+SC, PHP+SC, and CoT+HM in terms of the probability they assign to the correct answer.
> The algorithm having the lowest (best) rank (i.e., rank 1) for a 'difficult' question has the highest probability on the `correct' answer (note that, this does not necessarily mean that the corresponding algorithm's output is correct).
> Similarly, the algorithm, which assigns the lowest probability on the 'correct' answer is ranked the worst (i.e., 3).
>
> **Continued in the next Official Comment**

---

> ### Author Response · Authors · 2024-11-27
> **Cont'd**
>
> **Our result:**
>
> Next, we count for each algorithm how many times it obtains rank 1, 2, and 3 on these 'difficult' questions and plot the  stacked-histograms of these ranks for all six datasets using the three GPT models in Figures 3-5.
>
> We observe that the proposed CoT+HM achieves the lowest rank based on the probability of correct answer across the 'difficult’ questions for all datasets and all LLMs more often, outperforming both CoT+SC and PHP+SC. The height of the blue bar in the CoT+HM column, which counts how many times CoT+HM has higher probability of the correct answer compared to either of the other two algorithms, is the largest for all datasets and LLMs in Figures 3-5.
>
> In other words, this provides **direct empirical evidence** that CoT+HM has **higher probability of the correct answer** compared to its competitors (including CoT+SC, which uses $p_1(\cdot|x)$ for inference) for most of these 'difficult' questions across six datasets and three LLMs (i.e., 18 different cases).
>
> Remember that for each question, obtaining a **higher probability on the right answer** from (one or more rounds of) CoT+HM than that of CoT+SC is possible **if and only if** the **"in-flow of probability to the correct answer is greater than out-flow of probability from the correct answer"**.
>
> We would like to stress that this experiment is not intended to serve as another "CoT+HM'' versus "CoT+SC'' competition, of the form "our proposed method works better''. The experiment evaluates the probability assigned to the correct answer, and this may not be the maximum, so it does not directly reflect the accuracy of a method. Rather, the value that "CoT+HM'' assigns to the correct answer $y$ is a direct empirical approximation of $p_3(\tilde{y}=y|x)$, and the value that "CoT+SC'' assigns to the correct answer $y$ is a direct empirical approximation of $p_1(\tilde{y}=y|x)$. When these approximations are formed using 40 chains-of-thought, they form a sufficiently accurate approximation of the underlying probabilities, such that when we perform a rank comparison over 6 datasets and 3 LLMs, the probability of observing such a consistent difference through chance is very small. Moreover, one could argue that in order for the proposed HM strategy to work, we actually need to observe the in-flow $>$ out-flow condition for the empirical probabilities.
>
> **Statistical Significance:**
>
> In order to demonstrate the statistical significance of our result, we have now conducted a Wilcoxon signed rank test between $p_3(y|x)$ (i.e., the estimated probability of the `correct' answer obtained from the proposed CoT+HM) and $p_1(y|x)$ (i.e., the probability of the 'correct' answer, at the initialization of CoT+HM, estimated from CoT+SC using 40 samples), and report the p-values in Table 11 in Appendix 8.9 in the revised version of the paper (also shown here).
> We observe that except for 5 out of 36 cases (6 datasets, 3 LLMs, and 2 different partitions of the datasets), the difference between $p_3(y|x)$ and $p_1(y|x)$ is statistically significant at the 5\% level, providing strong empirical support in favor of the capability of the HM iterations in increasing the probability of the true answers.
>
> In addition, we also calculate the percentage of difficult questions for which $p_3(y|x) \geqslant p_1(y|x)$ is satisfied and report the results in Table 12 in Appendix 8.9 in the revised version of the paper (also shown here).
> We observe that in each case, for the majority of the questions, HM iterations do not decrease the probability of the true answer.
>
> To comply with the reviewer's suggestion, we have included these results in Appendix 8.9 to demonstrate the empirical capability of the hint prompt in refining the probability distribution of answers.
>
> **Other comments:**
>
> Note that we **do not claim** that **in-flow of probability to the correct answer is greater than out-flow of probability from the correct answer** for **all questions in all datasets**.
>
> Since on the 'easy' questions as defined above, both CoT+SC and CoT+HM have 100\% accuracy, in order to obtain an improved performance using CoT+HM, we only need a subset, consisting of 'difficult' questions, where the property is satisfied more often, and this increase in probability of the 'correct' answer with each round of HM results in a correction of the final answer.
>
>
> **Summary:**
>
> 1) Our results in Figures 3-5 substantiate that we indeed observe the increase in probability of the correct answer more often by applying HM using the hint prompt of Zheng et al., 2023.
>
> 2) As per the reviewer's suggestion, we provide additional details to clarify the results in Figures 3-5 here and modify Appendix 8.9 to include some quantitative results (Table 11 and 12) to justify the use of hint-prompt in our work.
>
>
> **Tables 11 and 12 in Appendix 8.9 are copied in the next Official Comment for completeness.**

---

> ### Author Response · Authors · 2024-11-27
> **Cont'd**
>
> **p-value from Wilcoxon signed rank test between the probabilities of true answers from distributions $p_3(y|x)$ and $p_1(y|x)$ for the 'difficult' questions (for the entire dataset)**
>
> | **LLM**              | **AddSub**                 | **MultiArith**            | **SingleEQ**             | **SVAMP**                | **GSM8K**                | **AQuA**                 |
> |----------------------|----------------------------|---------------------------|--------------------------|--------------------------|--------------------------|--------------------------|
> | **GPT-3.5 Turbo**    | 0.0291 (0.1172)            | 0.0006 ($1.3 \times 10^{-5}$) | 0.0012 ($8.6 \times 10^{-5}$) | 0.0132 ($1.4 \times 10^{-5}$) | $9.2 \times 10^{-18}$ ($4.3 \times 10^{-22}$) | 0.0001 ($1.6 \times 10^{-8}$) |
> | **GPT-4 Turbo**      | 0.2868 (0.2258)            | 0.0104 ($2.3 \times 10^{-6}$) | 0.0002 ($6.2 \times 10^{-7}$) | $4.8 \times 10^{-8}$ ($1.7 \times 10^{-13}$) | $2.2 \times 10^{-31}$ ($1.5 \times 10^{-41}$) | 0.0065 (0.0042) |
> | **GPT-4o-mini**      | 0.0038 (0.0024)            | 0.8413 (0.0243)            | 0.0317 (0.0255)           | 0.5898 (0.3028)          | $4.5 \times 10^{-12}$ ($5.2 \times 10^{-12}$) | $2.1 \times 10^{-5}$ ($8.5 \times 10^{-6}$) |
>
> **Percentage of 'difficult' questions (percentage of questions in the entire dataset), so that $p_3(y|x) \geqslant p_1(y|x)$ is satisfied (in other words, HM does not decrease the probability of the true answer)**
>
> | **LLM**              | **AddSub**       | **MultiArith**    | **SingleEQ**      | **SVAMP**         | **GSM8K**         | **AQuA**          |
> |----------------------|------------------|-------------------|-------------------|-------------------|-------------------|-------------------|
> | **GPT-3.5 Turbo**    | 79.4 (92.7)      | 85.2 (97.3)       | 86.0 (97.2)       | 63.5 (83.8)       | 70.8 (81.4)       | 64.7 (74.8)       |
> | **GPT-4 Turbo**      | 76.2 (95.7)      | 96.3 (99.7)       | 87.7 (98.0)       | 89.5 (96.9)       | 85.7 (93.3)       | 79.1 (86.6)       |
> | **GPT-4o-mini**      | 85.7 (97.2)      | 96.3 (99.7)       | 82.5 (97.0)       | 81.1 (93.9)       | 83.8 (92.7)       | 75.8 (83.9)       |
>
>
> > # A priori, I find the prompting strategy you adopt (from the PHP work) somewhat strange, especially given that the few-shot rationales don't appear to reference the hints. I understand (and appreciate) your arguments for why this style of hinting could help in quantitative domains (maybe somehow attending to the hint makes nearby answers more likely), but still, I find that explanation far from obvious and in need of empirical validation.
>
> Please refer to the answer above for the detailed empirical validation of how PHP-style prompting helps in instantiating our HM framework.
>
> While we agree that, to some extent, the design of the PHP prompt is not completely intuitive, and the lack of extensive analysis and ablation is a valid criticism of (Zheng et al., 2023)'s work, we argue that those criticisms should not overshadow the novel methodological contributions made in our work because of the following reasons.
>
> First, empirical results in both (Zheng et al., 2023) and our submitted work clearly show that **PHP outperforms CoT**, which **provides evidence in favor of the usefulness of hinting.**
>
> Second, the utility of our design of HM framework is motivated by a principled criterion that **'probability of the correct answer increases via hint-marginalization'** if and only if **'the in-flow probability to the correct answer is more than the out-flow probability from the correct answer'**. Our empirical analysis (Figures 3-5, please refer to the detailed response above) clearly shows that there is strong empirical evidence in support of that phenomenon for the hinting prompt across all datasets and LLMs considered in our experiments. **Thus, the utilization of the PHP-style prompt as a component in our work is well justified.**
>
> We agree with the reviewer that a deeper investigation into why the hinting approach is beneficial would be a valuable contribution, but we do not think it is essential to fully understand why a mechanism works provided there is convincing evidence that it does work.
>
> **Continued on the next Official Comment**

---

> ### Author Response · Authors · 2024-11-27
> **Cont'd**
>
> > # Which brings me to Appendix 8.3--thank you for pointing me to this experiment. However, I am unsure how best to understand the results. (This is an average across the whole dataset? Did you query GPT-4 Turbo more than once per prompt, or only once per prompt?)
>
> In Appendix 8.3, we have written "As an example, using GPT-4 Turbo on the entire GSM8K dataset, the empirical frequency of obtaining an incorrect answer conditioned on an immediate correct hint is 0.0179.
> This suggests that assuming $\gamma$ to be very small is justified.
> On the other hand, the empirical frequency of obtaining a correct answer conditioned on a previous incorrect hint is 0.3159, which supports the assumption of having a strictly non-zero value for $\delta$.''
>
> (1) These results are obtained by considering an average across the whole dataset. We have amended the text in the appendix to make this clear.
>
> (2) We queried the LLM more more than once per prompt, since these results are obtained by analyzing PHP+SC.
>
> The original purpose of Appendix 8.3 was to convey the simple intuition for Section 3.1, that on average, LLMs tend to repeat the correct answer with a high probability if it is provided as a hint, and they also possess some ability of self-correction if an incorrect hint is provided.
>
> However, in light of the reviewer's questions, we realize that reporting a dataset level summary statistic only provides some indirect evidence in support of the success of the hinting mechanism.
>
> On the other hand, we already presented a more thorough, direct, and systematic study of PHP prompt's refinement capability in the paper (as explained in detail in the response to the previous question above, with results in Figures 3-5).
> We have also added quantitative results from the same experiment to the modified Appendix 8.9 which concretely show support for our usage of the hint-prompt in the HM framework.
>
> **Continued in the next Official Comment**

---

> ### Author Response · Authors · 2024-11-27
> **Cont'd**
>
> > # The numbers you report seem consistent with your assumption, but also with other possibilities, e.g. that a hinted LM has some probability $q$ of sticking with the hint, and probability $1-q$ of ignoring the hint and answering from its unhinted distribution. Because GPT-4 Turbo already does very well on the dataset, this will look like "high probability of correct given correct hint, some probability of correct given incorrect hint." But on a dataset where the initial performance was worse, this behavior might no longer satisfy your assumptions.
>
> We thank the reviewer for this insightful question. In light of your analysis, we realize that the provided results in Section 8.3 do not act as sufficient support for the 'in-flow vs out-flow' assumption.
>
> As mentioned in the previous response, the original purpose of Appendix 8.3 was to provide some evidence in support of our assumptions in Section 3.1.
>
> Please refer to the **previous response and the updated Appendix 8.9 for a detailed discussion of this issue**.
>
> However, **while the proposed hypothetical scenario could explain the numbers presented in Appendix 8.3, it cannot explain the observations in Figures 3-5**, as explained below.
>
> Let us first consider two extreme scenarios: a) the hint conditioned LLM outputs the hint as answer with certainty, i.e., $p(y_1|x, \mathrm{Hint}(y_2)) = \delta(y_1-y_2)$ for all $y_1$ and $y_2$ ($\delta(\cdot)$ denotes the Kronecker delta function), and b) the hint conditioned LLM ignores the hint completely and answers from its unhinted distribution $p(y_1|x, \mathrm{Hint}(y_2)) = p_1(y_1|x)$ (note that $p_1(\cdot|x)$ is the initial (unhinted) distribution, estimated using CoT+SC) for all $y_1$ and $y_2$.
>
> In case a),
> \begin{align}
> p_{2}(\tilde{y}|x) &= \int p(\tilde{y}|x, \mathrm{Hint}(y')) p_{1}(y'|x)dy' = \int \delta(\tilde{y}-y') p_{1}(y'|x)dy' = p_{1}(\tilde{y}|x)
> \end{align}
>
> In case b),
> \begin{align}
> p_{2}(\tilde{y}|x) &= \int p(\tilde{y}|x, \mathrm{Hint}(y')) p_{1}(y'|x)dy' = \int p_1(\tilde{y}|x) p_{1}(y'|x)dy' =  p_1(\tilde{y}|x) \int p_{1}(y'|x)dy' =  p_1(\tilde{y}|x)
> \end{align}
>
> So, in both cases a) and b), HM would keep the answer distribution unaltered.
> Intuitively, if the LLM outputs the hint as its answer w.p. 1, then both the in-flow and out-flow probabilities are zero.
> On the other hand, if it ignores the hint completely, the in-flow and out-flow probabilities are equal.
>
> The reviewer's hypothetical scenario is a mixture of those two extreme conditional distributions, i.e., ``a hinted LM has some probability $q$
>  of sticking with the hint, and $1-q$ probability
>  of ignoring the hint and answering from its unhinted distribution''.
>  In this case, we have $p(y_1|x, \mathrm{Hint}(y_2)) = q \delta(y_1-y_2) + (1-q) p_1(y_1|x)$.
> From the analysis above, we see that for any value of $q \in [0,1]$, this would again result in $p_{2}(\tilde{y}|x) =p_{1}(\tilde{y}|x)$ for all $\tilde{y}$.
>
> **So, if the reviewer's hypothetical scenario is indeed true, then HM would not increase the probability of the correct answer (and hence would not increase accuracy), irrespective of the value of the initial accuracy obtained by estimating mode of the unhinted distribution $p_1$.**
>
> However, our results show that a) **CoT+HM outperforms CoT+SC** (which outputs the mode of $p_1$ as its answer) in Table 1, and b) more often **CoT+HM has higher probability of the correct answer than that of CoT+SC** across different datasets and LLMs (Figures 3-5).
>
> Those observations **support our intuitions about the usefulness of hinting**. Other results in our paper (e.g., the **comparison between CoT and PHP in Table 1**) also provide **strong empirical evidence** that the **PHP-style prompt is indeed taking the hint into account in a positive way**.
>
> > # Thank you for your various other clarifications. I apologize for the language "essentially presenting a new prompting technique" -- it's true you are not really presenting a prompting technique, more an inference-time iterative prompting protocol. I do think the idea makes sense, but still have reservations about "hinting." I note Reviewer mTdx had similar concerns. I can see arguments for accepting this paper but on balance I am still somewhat dissatisfied. I am fine conceding to other reviewers with stronger opinions or more background in this area, though, if they are convinced by the results.
>
> Thank you for acknowledging that we "are not really presenting a prompting technique'' and the idea "makes sense''.
>
> **We very much appreciate the interaction and the careful thought you have given to our work; it has prompted us to consider some important aspects more rigorously.**
>
> Please let us know if you still have any outstanding concerns, so that we can make further attempts in addressing them.

---

### Official Review · Reviewer_v83X · 2024-11-05

**Soundness:** 3
**Presentation:** 3
**Contribution:** 2
**Rating:** 6
**Confidence:** 2

**Summary:**

The paper presents a new method called hint marginalisation to improve the reasoning capability of a large language model. The basic idea is to generate multiple responses (possibly in parallel) from the same model or from multiple models and combine in order to steer the model towards the most likely response. Therefore, the proposed approach can be viewed as an iterative sampling scheme to produce a Monte-Carlo approximation of a probability distribution over the responses where the mode of the distribution corresponds to the most likely response. The empirical evaluation is carried out on several reasoning tasks using OpenAI GPT models. The results demonstrate conclusively that the proposed hint marginalisation scheme improves the reasoning capabilities of the models considered.

**Strengths:**

- Improving the reasoning capabilities of existing language models is definitely a topic of considerable interest in the AI community. The proposed approach seems to address this issue in a principled manner.

- The quality of the presentation is overall quite good and therefore the paper is relatively easy to follow even by readers outside this research area. Most of the technical details presented in the paper are discussed in a relatively clear manner. The examples provided throughout the paper help to get a better understanding of the proposed scheme.

**Weaknesses:**

- Most modern LLMs, especially those from the OpenAI GPT family do fairly well on arithmetic reasoning problems and therefore the improvements shown in Table 1 are marginal (typically less than 1%). Perhaps considering other reasoning tasks would better highlight the benefits of the proposed approach.

- I was surprised to see that only  the GPT models were considered in the experimental evaluation. They already demonstrated strong reasoning capabilities. Therefore, maybe the proposed approach would be more appropriate for weaker models.

**Questions:**

- Can you comment on applying the hint marginalisation scheme to open models like the Llama family? And if you already experimented with the open models, what kind of results did you obtain?

---

> ### Author Response · Authors · 2024-11-24
> **Response to Reviewer v83X**
>
> We thank the reviewer for acknowledging the principled nature and clear presentation of our work. Below, we address your concerns.
>
> > #  W1.  Most modern LLMs, especially those from the OpenAI GPT family do fairly well on arithmetic reasoning problems and therefore the improvements shown in Table 1 are marginal (typically less than 1\%). Perhaps considering other reasoning tasks would better highlight the benefits of the proposed approach.
>
> We thank the reviewer for this suggestion.
> We agree that some of the arithmetic  datasets are relatively easy for GPT, but we still observe consistent improvement over a strong baseline (self-consistency) in 15 out of 18 cases in Table 1.
> The same benchmarks are considered in the papers that presented the relevant baselines.
> Our proposed hint marginalization strategy solves problems where self-consistency fails to establish the correct mode (please refer to Figure 2 in our paper) and sampling more CoTs is not helpful. If we restrict ourselves to the 'difficult' questions during the performance assessment (eliminating the easy questions that are answered correctly by all LLMs and all methods), then the improvement is more substantial (please refer to Table 10).
>
> The lack of particularly challenging datasets is a valid criticism of our work, also raised by other reviewers.
> We have now included results for the MATH dataset (please refer to Table 8), which is a much more challenging mathematical reasoning dataset. For several sub-disciplines (Geometry, Intermediate algebra, Pre-calculus), the state-of-the-art performance (without using extreme computation and a very long inference time) is in the range of 50-65 percent, suggesting that LLMs still find these problems very difficult to solve. The proposed HM approach leads to a performance improvement in 5 out of 7 settings.
>
> We agree with the reviewer that applying the method more broadly to
> other reasoning domains is a worthwhile and very interesting research direction.
> With a view to satisfying this request, we now provide results for "Date Understanding" and "Object Tracking", which are problems sets involving quantitative (but not strictly mathematical or arithmetic) reasoning (please refer to Table 9 for results).
>
> Extending beyond this (outside quantitative problems) would require careful prompt engineering to generalize hinting to other reasoning domains. This direction is very interesting but is not the main focus of our current work.
>
> > # W2. I was surprised to see that only the GPT models were considered in the experimental evaluation. They already demonstrated strong reasoning capabilities. Therefore, maybe the proposed approach would be more appropriate for weaker models.
>
> This is a good suggestion. We agree that it is important to extend analysis beyond the GPT family. We now include results for two Llama models (please refer to Table 7 for results).
>
> > # Q1. Can you comment on applying the hint marginalisation scheme to open models like the Llama family? And if you already experimented with the open models, what kind of results did you obtain?
>
> We have now conducted experiments with two Llama-family LLMs: the weaker Llama-3-8b-instruct and the very capable Llama-3-70b-instruct.
> The results are presented in Table 7.
> In order to reduce the API cost of the experiments, we restrict running the more expensive 70B model to only the three most difficult benchmarks.
>
> From the results in Table 7 of our revised paper, we observe that using Llama-3-8b-instruct, the relative advantage of PHP over CoT is diminished in comparison to the GPT models.
> This suggests that weaker LLMs, such as Llama-3-8b-instruct, which often have relatively poor instruction following capability, cannot utilize the hint effectively for solving the reasoning task, highlighting the inadequacy of sophisticated prompting for weaker LLMs.
>
> In this setting, the effect of the quality of approximation of the initial distribution of HM becomes important for obtaining a good reasoning accuracy and PHP+HM outperforms CoT+HM in most cases.
> Except for GSM-8K, PHP+HM either outperforms CoT+SC or obtains comparable performance on all other datasets.
>
> On the contrary, for a strongly capable Llama-3-70b-instruct model, both CoT+HM and PHP+HM perform well.

---

> ### Author Response · Authors · 2024-11-27
> **Please let us know if you still have any outstanding concerns**
>
> Dear reviewer v83X,
>
> Thank you for your review and the acknowledgment of our response.
>
> As the discussion period nears its end, we hope that we have effectively addressed and resolved your concerns.
> Please let us know if you still have any outstanding concerns, so that we can make further attempts in addressing them.
>
> In response to your comments, we have performed additional experiments using two Llama family models and showed the application of HM beyond arithmetic tasks (Math dataset and two big-bench tasks).
> In both cases, we obtained accuracy improvement using the proposed HM framework.
>
> We believe that these new experiments address your core concerns and therefore would like to request a reconsideration of your rating of our paper.

---

### Author Response · Authors · 2024-11-24
**General response to all reviewers**

We would like to sincerely thank the reviewers for their thoughtful and constructive feedback on our paper. Their comments have  helped significantly in improving the quality of our work. We deeply appreciate the time and effort that each reviewer invested in thoroughly reviewing our manuscript. The detailed suggestions and observations were invaluable, and we believe that the revisions made in response to their comments have strengthened the paper considerably.

As per the reviewers' suggestions, we have added a) results using Llama-3 (Table 7), b) results on Math dataset (Table 8), c) results on tasks beyond arithmetic reasoning (Table 10), and d) discussion on the intuition of using hints (Appendix 8.8) in the revised version of the paper.

We would like to share these new results with all reviewers. Below, we respond to each reviewer individually.

---

> ### Author Response · Authors · 2024-11-28
> **Summary of new experimental results obtained during the rebuttal period**
>
> For the sake of the reviewers' convenience, we provide a **brief summary of the new experimental results we obtained during the rebuttal period.**
>
> **Results using Llama:**
>
>  From the results in Table 7 in our revised paper, we observe that for a strongly capable Llama-3-70b-instruct model, both CoT+HM and PHP+HM perform well and outperform CoT+SC. The results for a strong Llama model thus align with those for the stronger GPT models.
>
> When using Llama-3-8b-instruct, PHP+HM algorithm achieves the best accuracy in 4 out 6 datasets and performs comparably in the remaining two datasets. For the weaker model, it is important to have a better initial distribution to refine, and PHP achieves this better than CoT.
>
> **Results on Math dataset:**
>
> Mean and standard error of accuracy (in \%) of reasoning on the Math dataset using GPT-4o-mini. The **highest** accuracy among all competing algorithms is marked in **bold** and the _second-best_ accuracy in those cases is marked in _italic_.
>
> | **Algorithm**     | **Algebra**  | **Counting and Probability** | **Geometry**  | **Intermediate Algebra** | **Number Theory** | **Prealgebra** | **Precalculus** |
> |-------------------|--------------|------------------------------|---------------|--------------------------|-------------------|----------------|-----------------|
> | **CoT**           | 88.5±0.9     | 73.4±2.0                     | 55.1±2.3      | 51.5±1.6                 | 76.3±1.8          | 86.9±1.1       | 49.1±2.1        |
> | **PHP**           | 90.2±0.9     | 75.3±2.0                     | 55.9±2.3      | 52.3±1.7                 | 78.1±1.8          | 87.6±1.1       | 51.1±2.1        |
> | **CoT+SC**        | 93.9±0.7     | **82.9±1.7**                 | *64.7±2.2*    | 58.1±1.7                 | *83.5±1.6*        | **91.2±1.0**   | 51.3±2.1        |
> | **CoT+HM**        | *94.1±0.7*   | *81.0±1.8*                   | 64.1±2.2      | *58.3±1.7*               | 82.0±1.7          | **91.2±1.0**   | *51.5±2.1*      |
> | **PHP+HM**        | **94.8±0.6** | 80.6±1.8                     | **65.3±2.2**  | **58.9±1.6**             | **85.4±1.5**      | *90.7±1.0*     | **52.0±2.1**    |
>
> We observe that **the HM approach leads to a performance improvement in 5 out of 7 settings.**
>
> **Other tasks:**
>
> Mean and standard error of accuracy (in \%) of reasoning for Date Understanding and Object Tracking tasks using GPT-4o-mini. The **highest** accuracy among all competing algorithms is marked in **bold** and the _second-best_ accuracy in those cases is marked in _italic_.
>
> | **Algorithm**     | **Date Understanding** | **Object Tracking** |
> |-------------------|------------------------|---------------------|
> | **CoT**           | 91.9±1.4               | 96.4±0.7            |
> | **PHP**           | 93.5±1.3               | *97.7±0.5*          |
> | **CoT+SC**        | *93.8±1.3*             | 96.7±0.7            |
> | **CoT+HM**        | **94.6±1.2**           | **98.0±0.5**        |
>
> We provide results for "Date Understanding" and "Object Tracking", which are problems sets involving quantitative (but not strictly mathematical or arithmetic) reasoning. We observe that PHP still outperforms CoT, demonstrating the utility of hinting beyond the arithmetic tasks. The proposed CoT+HM offers an improvement in accuracy for both of these datasets by reducing the average error rate by more than 10 percent compared to the next best baseline.

---

### Meta-Review · Area_Chair_5oqG · 2024-12-19

**Metareview:**

This paper generated a lot of discussion. The general opinion was that the paper identified an important problem of combining multiple responses from LLMs to improve their reasoning ability. The proposed approach was also considered as principled.

While the broad ideas were well articulated, the reviewers were quite concerned with what they perceive as ad-hoc evaluation. It is not clear why the improvement shrunk compared to benchmarks, and while a small experiment was added for discrete arithmetic, no methodology was provided, analysis of ranks and p-values were provided for only some subsets of benchmarks and Llama was evaluated on a subset of benchmarks with no rank histograms.

In the end, there was consensus that the paper needs one more round of edits before acceptance. IF this was a journal, I would have suggested major revisions and then reviewed it but the changes do not appear minor.

**Additional Comments On Reviewer Discussion:**

The reviewers were actually quite responsive and engaged with the authors. There was some resentment among the reviewers with the tone of the discussion from the authors. I would request the authors to kindly be a bit more polite with the responses if possible.

---

### Decision · Program_Chairs · 2025-01-22

Reject